



# Ideas and Perspectives:
# When ocean acidification experiments are not the same, reproducibility is not tested

Phillip Williamson[1], Hans-Otto Pörtner[2], Steve Widdicombe[3], Jean-Pierre Gattuso[4,5]

[1]School of Environmental Sciences, University of East Anglia, Norwich NR4 7TJ, UK

[2]Alfred Wegener Institute for Polar and Marine Research, 27515 Bremerhaven, Germany

[3]Plymouth Marine Laboratory, Plymouth, PL1 3DH, UK

[4]Sorbonne Université, CNRS, 06230 Villefranche-sur-mer, France

[5]Institute for Sustainable Development and International Relations, 75006 Paris, France

*Correspondence to*: Phillip Williamson (p.williamson@uea.ac.uk)

**Abstract**

Can experimental studies on the impacts of ocean acidification be trusted? That question was raised in early 2020 when a high-profile paper failed to corroborate previously-observed impacts of high $CO_2$ on the behaviour of coral reef fish. New information on the methodologies used in the 'replicated' studies now provides the explanation: the experimental conditions

were substantially different. High sensitivity to test conditions is characteristic of ocean acidification research; such response variability shows that effects are complex, interacting with many other factors. Open-minded assessment of all research results, both negative and positive, remains the best way to develop process-based understanding of those responses. Whilst replication studies can provide valuable insights and challenges, they can unfortunately also be counter-productive to scientific advancement if carried out in a spirit of confrontation rather than collaboration.


## 1. Introduction

Ocean acidification refers to a reduction in seawater pH (increased hydrogen ion concentration) over an extended period, typically decades or longer, caused primarily by the uptake of carbon dioxide ($CO_2$) from the atmosphere. Associated chemical changes include increased concentration of bicarbonate ions and dissolved inorganic carbon, and reduction in the

concentration of carbonate ions in the ocean and, unless compensated for, the body fluids of marine organisms. Although the chemistry of the carbonate system has been very well understood for decades, research on the biological and ecological implications of anthropogenic ocean acidification only began in earnest about 20 years ago (Gattuso and Hansson, 2011). A wide range of consequences have been identified, with differential vulnerability of key animal phyla (Kroeker et al., 2013; Wittmann and Pörtner, 2013). The production of shells and skeletons is considered most sensitive to ocean acidification;

however, there is also strong evidence for low pH conditions affecting animal behaviour (Clements et al., 2015).



## 2. The challenge of contradictory results

A two-step experiment has been used by many research groups to investigate the possible effects of ocean acidification on fish behaviour. Initially, individual fish are given a binary choice of water conditions in a flume tank, with one choice

including an odour (e.g. from predators) known to elicit an avoidance response.  Those observations of discriminatory ability then provide the 'control' strength of preference, for comparison with treatment results using the same choice under raised $CO_2$ (lowered pH) conditions throughout the test tank.

Based on that approach, Clark et al. (2020a) reported their findings in an unambiguously-titled paper: "Ocean acidification

does not impair the behaviour of coral reef fishes". To exclude the possibility of inadvertent observer bias, they deployed video recording and automatic tracking software in their study, making that digital information openly available. They also used data simulations to conclude that other research groups' results were 'highly improbable', with an estimated likelihood of 0 out of 10,000 − assuming identical experimental conditions and that their own data were valid.  Since Clark et al. went to 'great lengths' to replicate earlier work yet failed to get the same results, there was a clear implication that other

researchers' work was unreliable, flawed or even fraudulent (Clark et al., 2017).

In the context of wider concern regarding research reproducibility and scientific integrity (Nature, 2018), Clark et al. (2020a) attracted media attention and scientific comment. In particular, Clark et al.'s apparent thoroughness was praised by advocates of replication studies and several independent commentators (Enserinck, 2020; Science Media Centre, 2020).

However, there was also early identification of three potential weaknesses. First, Clark et al. (2020a) *did* find several significant ocean acidification effects, contrary to the paper's title, although less dramatic than those previously reported. Second, their analysis gave scant attention to the extensive literature on factors causing variability in ocean acidification research. The third, more fundamental, concern related to how closely the original experiments had been repeated, and whether that issue had been thoroughly checked during the paper's peer review.


## 3. Experimental differences

Any deficiencies in the original scrutiny of Clark et al. (2020a) have now been remedied, with the (online) publication of a detailed critique by Munday et al. (2020a), providing comprehensive evidence that the experimental conditions differed in many important ways: "*Clark et al. did not closely repeat previous studies, as they did not replicate key species, used*

*different life stages and ecological histories, and changed methods in important ways that reduce the likelihood of detecting the effects of ocean acidification.*"

Experimental differences between the original and repeated results included the following:
- Clark et al. (2020a) did not use clownfish, one of the original test species



- 65    ● Adult and sub-adult fish were mostly used, rather than larvae and small juveniles (with older fish known to be less responsive to risk cues)

     ● For one species, the juveniles were from an inbred aquarium population (likely to be pre-adapted to high $CO_2$, and hence less sensitive)

     ● Many experiments were carried out during a marine heatwave (with high temperatures known to reduce or reverse
- 70      responses in the studied species)

     ● Dissolved $CO_2$ levels were unstable (average daily $pCO_2$ range of 581 μatm in 2016 treatments), not meeting international standards for ocean acidification research (Riebesell et al., 2011).

There were also crucial changes to the design of testing apparatus, the dilution and nature of odour cues, and duration of tests. Such changes weakened the control response, hence reducing the likelihood of significant $CO_2$ treatment effects. In

total, 16 differences between the original studies and the re-runs were identified by Munday et al. (2020a), any one of which could invalidate the comparisons.

The counter-argument, made at the time of the original publication (Enserink, 2020) and subsequently re-iterated by Clark et al. (2020b), is that minor experimental differences are inevitable, and can be considered as reflecting natural environmental

variability. They should not matter if the original findings are widely applicable and robust. The question of what does or does not constitute a valid replication is therefore critical, yet inherently problematic. Whilst accepting that an exact repeat of a biological study is impossible ("*No man ever steps in the same river twice…*"; Heraclitus), it would seem useful to distinguish 'reproducibility' from 'replicability' – with a test of the former requiring that an experiment should be repeated as closely as possible, e.g. with the same experimental set-up, whilst greater flexibility is allowed for the latter, e.g. defining

replication as "*a study for which any outcome would be considered diagnostic evidence about a claim from prior research*" (Nosek & Errington, 2020a). Indeed, Clark et al. (2020a) also use the term 'conceptual replication' in their supplementary material, acknowledging that not all experimental conditions will be the same.

Recognition of the limitations of replication needs to be consistent to avoid contributing to the confusion. There are two

further issues. First, it is important that attempted replications do not ignore key components of the original hypotheses; for example, by neglecting the life-stage dependence of the response. Second, broadening the concept has a clear corollary: scientific controversies that may arise cannot be resolved by any single experiment, but need to take into consideration all relevant studies, with awareness that there were differences between them. A total of 110 research papers have investigated how ocean acidification might, or might not, affect the sensory physiology of fish, as identified in Supplementary Table 1 of

Munday et al. (2020a). Out of 44 that involved coral reef fish, 41 of those studies, carried out by 68 researchers at 35 institutions in 15 countries, reporting significant effects, including several that used video recording, blind-testing, and raw-data publication.



The outcome for the other 66 papers (for other tropical, temperate and polar fish; marine and freshwater) provided additional support: 44 of those reported significant behavioural effects of ocean acidification. A closely similar result was found in a meta-analysis of 95 marine and freshwater studies by Clements et al (2020), with T.D. Clark included in the authorship team: they found that 64 of those papers reported either strong or weak effects.  Whilst the proportion showing a strong effect declined over the period 2009 -2019, that decrease could be expected, since the early strong effect studies were all on the

most sensitive (marine) species.  Further independent evidence is provided by molecular studies, showing direct effects of high $CO_2$ on neurotransmission in fish (e.g. Schunter et al, 2019) and a wide range of other taxa (e.g. Moya et al., 2016); additional biochemical and pharmacological examples are given by Munday et al (2020a).  It can therefore be confidently concluded that ocean acidification has the potential for adverse impacts on fish behaviour, whilst also recognising that such impacts do not occur under all circumstances and that some species may be resilient.


**4. Taking account of response variability**

Global environmental change has impacts on marine organisms and ecosystems which can affect ecosystem services and society (Bindoff et al., 2019). A good understanding of these responses is crucial to guide mitigation and adaptation.  Our increasing appreciation of the complexity of biochemical, physiological, behavioural and ecological interactions with ocean

acidification is both scientifically exciting and sobering, showing the difficulty in developing comprehensive understanding of the impacts of this important component of ocean climate change. But those complications should not be surprising, since natural variability in pH and $CO_2$ levels will have been experienced by marine species throughout their evolution. Species will inevitably have found different ways of responding, under many different environmental conditions, and such response differences can also be expected to occur in experimental studies.


Response variability in ocean acidification experiments is not novel. It was noted for studies on survival, calcification, growth and reproduction in early meta-analyses (Kroeker et al., 2013), and has subsequently provided the focus for much national and international research.  It is therefore now well-established not only that closely-related marine species can respond very differently to experimental pH treatments, but also that the magnitude of single species responses can be

significantly affected by many factors, including length of exposure, population-level genetic differences due to local adaptation, food availability, interactions with other stressors, seasonality, energy partitioning and the sex of the organisms used in experiments (e.g. Thomsen et al., 2012; Suckling et al, 2014; Sunday et al., 2014; Breitburg et al, 2015; Vargas et al., 2017; Ellis et al., 2017), as well as physico-chemical conditions (Riebesell et al., 2011).

It is therefore over-simplistic and unscientific to consider that results from any single ocean acidification study provide the final word, over-riding other findings in that topic area.  Instead, the resolution of outstanding knowledge gaps requires fair assessment of all the available evidence, with further effort given to novel experimental design in a collaborative,



comparative framework (Boyd et al., 2018). In that context, replication studies and the reporting of negative results (Browman, 2016) both have important roles in testing the wider applicability of observed impacts, teasing out what

additional factors may be important in determining the scale and occurrence of experimental effects. However, to avoid the unsatisfactory sequence of unjustified initial criticisms (Clark et al 2020a), rebuttal (Munday et al 2020a), reply (Clarke et al.2020b) and a further point-by-point response (Munday et al. 2020b), those parties whose work is being repeated need to be involved in planning replications, with a more nuanced, non-confrontational framework for analysis and interpretation (Fanelli, 2018; Nosek and Errington, 2020a,b).


**Author contributions:** PW prepared the original draft with subsequent input and editing by all co-authors.

**Competing interests:** The authors declare that they have no conflicts of interest.

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
