# Peer review of "When ocean acidification experiments are not the same, reproducibility is not tested"

_Biogeosciences, 2020_

## Short Comment (SC1) · 11 Nov 2020

Some observations that might help improve the narrative:

1. The opinion reinforces the argument about the need for proper experimental design, both within and across experiments, but doesn't explicitly provide solutions or minimal requirements. I have emailed you the recent Haddaway perspective (https://doi.org/10.1038/s41559-020-01295-x), maybe some of the styling from that would be good, but either a Boxed flow diagram or take home message at the end of each section of what readers should do would be good.

[Figure]

2. Three things are missing for me. (i) one glove doesn't fit all, and what is high variability in one system (e.g. pelagic) is within the noise in another system (e.g. benthic), and that needs to be recognized, especially in review. Variability in carbonate chemistry between systems has not been summarised anywhere, although could be now that there are so many papers, and then these need to be couched within temporal variation (e.g. diurnal) for the same system. (ii) the detection of treatment effects is valid, even if the carbonate chemistry is different to other experiments. You hint at this, but this would benefit from some elaboration. (iii) reading most ocean acidification papers, the narrative is that OA is 'bad' which is not necessarily true. I take the point that calcifiers are affected, but would challenge the statement that they are most sensitive as this is only one parameter and there is a bias in the literature (people have picked calcifiers). You highlight other examples, including behaviour, in non-calcifying species which could be equally devastating to that species. I key message is that this literature base needs to move on from documenting effects, and think about what the consequences of those effects are for species interactions, fitness, reproduction/growth and the rest of the foodweb etc etc.

3. is there a standard checklist, or could you provide one in this article, of what authors should be reporting in every paper, e.g. in a table in supplementary? The carbonate chemistry, but what else? Which of these are necessary, and which are nice to have?

4. It would be good if you could add some commentary about being pragmatic. Alkalinity, in particular, is expensive so an experiment with hundreds of replicates cannot hope to achieve regular samples from all units on a daily basis. You need some, but there are ways to achieve something sensible (e.g. a set number of aquaria within each treatment, once a week or something). There has to be some common sense, but also some indication of what the acceptable minimum is. The point I am making, and have witnessed at several OA meetings, is that the conversation about chemistry can go way further than is needed when discussing accuracy, repeatability and reproducibility. All of these depend on the system you are in, what the question you are asking

is, and what is practically possible, i.e. the requirements are context dependent. A related point is that what works in one system should not dictate what is acceptable in another system. An analogy is the US water quality standards – when they were brought in some states were automatically above threshold as the ground composition was markedly different to the areas where the standards were formulated.

5. Variability – there are more sophisticated statistical methods available to look at variability and outliers. There is a danger that trying to make everyone confirm to a very controlled set of conditions means that you lose the insights from the variability that you have factored out. Part of the answer has to be embracing variability and using appropriate statistical approaches to account for and/or explore them. Meta-analysis is one way to look at multiple experiments, but not the only way and much could be done with mixed modelling, GLMM, GAMM and then specific analyses that analyse outliers (rather than account for them).

Hope this helps

---

## Short Comment (SC2) · 11 Nov 2020

This manuscript highlights important messages:

- only a combination of approaches can provide an answer to a complex question such as the biological impact of ocean Acidification

- no experiment can fully capture the complexity of the problem and fully replicate an experiment is anyway impossible.

More importantly, it addresses the importance of constructive discussion in the imperfect world of experimental biology.

[Figure]

I have a few suggestions:

- I would tone down the first sentence of the Abstract. I do not think that Clark et al. is addressing such a general question. They were rather focusing on the work by Munday and colleagues.

- I would make the point that NO ocean acidification experiment can anyway simulate what is happening in the real world. So nobody is completely right or wrong if the experiment is well conducted (no fraud or big flaws) and honest with its limitations. You cannot design an experiment that will include the complexity of the real work and the time scale. This is another argument for a combination of experiments using different approaches (to answer different questions and then different part of the puzzle).

- The manuscript focusses on the Munday vs. Clark et al. recent disagreement. Maybe it would be beneficial to can also include another example. A similar discussion on the relevance of experimental studies occurred after the publication of the paper of Cornwall & Hurd (2015) highlighting the use of suboptimal experimental practices in many published articles in the field of ocean acidification. The article was covered by a short text in Nature entitled: 'Seawater studies come up short' (Nature 524, 18–19; 2015) and followed by several media articles concluding that ocean acidification research was flawed. In a short response to Nature ("Laboratory seawater studies are justified", Hurd et al. 2015), we made similar points as this manuscript and highlighted the fact that laboratory studies are not ideal but one of the many tools (together with paleo studies, field work, models) that allow to capture the biological impact of ocean acidification.

---

## Short Comment (SC3) · 11 Nov 2020

This type of comment fosters discussions that we need to continue within science in general, and especially within global change biology research. As the field continues to evolve, what has struck me the most has been the variability in responses at many scales. In this commentary, Williamson et al. raise a valid point that looking at consensus of many studies in the field, particularly when they can encompass a variety of regional or life history based trends, should be the basis of drawing major conclusions. At the same time, the validity of single studies - in this case, whether Clark et al. 2020 or the studies under question therein - remains important in relation to that

consensus. What does it mean for a study to have a different or controversial result? Typically, this means that as a community of scientists we do not fully understand the mechanisms or processes at play. In this sense, the different results obtained given several small changes in methodology should be used to identify which mechanisms warrant further study. In science and in climate change biology especially, the book is not always closed.

Students especially will benefit from interactive discussions with global experts on issues of reproducibility but also learning how scientific consensus is formed and what is required for it to be overturned. How should one weigh the results of a single study in relation to the consensus of the field? How might questions and interpretations arising from a single study impact the direction of a field, or mechanisms to focus on in future study, relative to consensus science? If one study has shown something interesting, what are the next steps to build consensus across systems and organisms?

This discussion is timely and it is to everyone's benefit that it continue. Recently, I completed training workshops for science communication with policy makers. The primary take-home of this workshop was that we must always present consensus science to non-scientists, and refrain from the temptation to share our dearest and most exciting new results. Just as points in our datasets, single studies build a confidence envelope around our consensus, but we cannot build a solid argument around single observations.

---

## Referee Comment (RC1) · Anonymous Referee #1 · 30 Nov 2020

General Comments:

The manuscript from Williamson et al. provides an important contribution to the discussion on the criteria that should be used in replication versus reproducibility studies, and the implications for each, using the recent discussion regarding the implications of ocean acidification on the behaviour of coral reef fishes as an example to tease apart this issue. I think that adding in a table or figure on this topic to help authors in the future to ensure that they meet these criteria in their studies would be helpful. Although I believe the authors' arguments are interesting and valuable, I think that the authors should consider editing the article in its entirety to prevent using subjective language

throughout. I think that this topic is important and the audience as a whole will be more receptive if the authors' suggested ideas are laid out objectively and without subjective criticism of the published work on both sides.

Specific comments:

Lines 17-19: Is this the authors' argument? That the replication study was "confrontational"? It seems rather subjective and less constructive than the remainder of the abstract. Rather, it would be useful to provide an assessment of how to interpret and take further actions to understand the discrepancy when they arise from different experimental conditions here to close out the abstract. Below, you provide some really interesting suggestions for criteria for categorizing replication vs reproducibility studies, and I think a sentence along those lines would be more useful here.

Line 22: Add in an "i.e." in front of "increased"

Line 26: Change "very well understood" to "high studied"

Line 35: Past studies have frequently used conspecific chemical alarm cues as the stimulus in these flume studies, and thus this stimulus should also be noted in the parentheses (such as in Welch et al. 2014 and Heuer et al. 2016).

Line 35: I would recommend adding some examples of references here that have used this methodology so that interested readers can further read studies that have employed this methodology.

Line 37: Again, provide some references here for examples when this methodology has been employed.

Line 39: For the purpose of the appearance of objectivity, I recommend removing the phrase "an unambiguously titled" and replacing it with the phrase "the paper titled". It allows the reader to draw their own conclusions about the Clark et al. 2020a paper's title from the argument that you present below.

[Figure]

Line 41: Replace "they" with "the authors"

Line 43: Add in the year following Clark et al. (Clark et al. 2020a)

Line 43.45: I recommend re-writing this statement in order to summarize the assertions in Clark et al. 2020a objectively. I recommend the following edit: "Since Clark et al. 2020 claim to have attempted to replicate the results obtained from earlier work without success, they imply that the earlier work was either unreliable, flawed, or fraudulent (Clark et al. 2017)."

Line 48: I recommend adding "substantial" here ("...attracted substantial media...")

Line 48: I again recommend adding the year here ("... the apparent thoroughness of the approach described in Clark et al. (2020a)...")

Line 50: Please be clear about WHO identified these potential weaknesses. Also, as detailed in the reply by Munday et al. 2020a Nature, there were actually many more criticisms than three, so you should be clear that there were SEVERAL criticisms, and that you are highlighting three in particular.

Line 63: I recommend editing to include the phrase "but not limited to" (as in, "...included (but not limited to) the following...")

Line 89: I recommend editing to say "...avoid creating confusion..." instead of "...avoid contributing to the confusion...". The recent Clark et al. 2020a paper created some confusion about the generality of CO2-induced effects on behaviour rather than contributing to existing confusion.

Line 90-91: I suggest editing this sentence to say: "First, it is important that replication studies examine key components of the original hypotheses, such as the life-stage dependence of the response to altered CO2 conditions."

Line 91-93: Your point in this sentence is not entirely clear. I believe you are trying to say that all available evidence must be considered when evaluating potential contro-

versies. Is that right? If so, please re-word so that aim is clear.

Line 94: For non-experts on this topic, provide some context for how studies of sensory physiology link to the studies on behaviour, which are the focus here.

Line 94: I believe that you mean Table 1 in Munday et al. 2020a, not supplementary table 1. Is that correct? If so, please correct this in the text.

Line 95-96: For the sake of readability, I recommend using parentheses instead of commas around the phrase summarizing the researchers, institutions, and countries involved.

Line 107: Add in a period after et al.

Line 107-109: I recommend rewriting this sentence to the following: "Given the plethora of independent evidence, ocean acidification likely does have adverse impacts on fish behaviour. However, the resilience of fishes to altered CO2 is likely to vary depending on the species and circumstances under investigation."

Line 112: Add in a comma between ecosystems and which

Line 113: Change good to strong

Line 114: Rewrite this sentence as follows: "Our increasing appreciation for the complex relationship between ocean acidification and the ocean's biochemical, physiological, behavioural and ecological interactions are both scientifically exciting and sobering..."

Line 116-119: These two sentences are somewhat difficult to understand. Can you re-write to improve their clarity and brevity? I believe that your point here is that the complexity of the relationship between ocean acidification and natural processes is to be expected given the variability that these species would have experienced throughout their evolutionary history. If so, just say that.

Line 130: The terms over-simplistic and unscientific are perhaps not the most constructive adjectives to use here. Consider changing this sentence to something like the following: "Given this known variability, the results from any one ocean acidification study will therefore be unable to overshadow all of the other previous findings from that area of research."

Line 131-132: I am not entirely clear what you are trying to assert with this sentence. My initial reading of this paragraph is that this sentence can be deleted in its entirety, to instead focus on how practices in the field can be improved generally, through more widespread publication of replication studies and negative results, which are notoriously difficult to publish and often end up in low impact journals, causing researchers not to prioritize dissemination of their important results.

Line 135-139: I am not entirely clear what you are suggesting here as an alternative approach. You suggest that the published back and forth between Clark et al. and Munday et al. is ineffective, but do not clearly layout an alternative framework for these sorts of controversies. What do you mean specifically by a "more nuanced, non-confrontational framework"? These specifics would be useful for taking constructive steps in the future.

---

## Referee Comment (RC2) · Anonymous Referee #2 · 18 Dec 2020

Review of "Ideas and Perspectives: When ocean acidification experiments are not the same, reproducibility is not tested" by Williamson et al.

This article summarizes the debate in the literature between Clark et al and Munday et al on the sensitivity of Australian coral-reef fish behavior to ocean acidification. It does not present new data; rather it weighs in on the process of doing science. I applaud the authors for taking up the debate between Clark et al and Munday et al. It is important for senior scientists that are not conflicted to engage given the high level of attention the Clark/Munday debate has gotten inside and outside of the academic community. Disagreement is healthy to science when done constructively; Williamson et al have

justly called out non-constructive behavior. I appreciate how Williamson et al framed their essay and that they confronted the controversy head on. That said, the content of the Williamson et al. essay could be richer, which would help it appeal to a larger portion of the scientific community. The essay currently omits important aspects of the scientific process that led to the situation described and lacks concrete suggestions for how to avoid similar situations. The authors might also more carefully examine the language they used to avoid participating in a "toxic" exchange.

1) This essay could strengthen its arguments by better incorporating the ideas presented in Nosek and Errington (2020). The most troubling step taken by Clark et al was that they elevated their findings to a level in which their work could singularly suggest a failure of the hypothesis that fish behavior is sensitive to carbon dioxide conditions instead of it being part of a broader constellation of research used to refine the "generalizability space" (term per Nosek and Errington) of the hypothesis. I argue that this difference between failure and generalizability should be addressed in Williamson et al as it pertains to maintaining good norms of conduct in the field. Failure of a single study to support a hypothesis is a learning opportunity, not a reason to cast doubt on the rigor of prior work. A similar line of thinking is presented in the response by McCoy and in an Oceanography article by Busch, O'Donnell et al 2015 (http://dx.doi.org/10.5670/oceanog.2015.29). As brought up in section 3 of Williamson et al, Clark et al's failure to properly account for the mechanistic explanations for observed effects of OA on fish behavior when interpreting their work is likely part of what led Clark et al to their conclusions. In doing so, they not only refute observation of the behavioral expression but the physiological work under-pinning it.

2) The essay omits a major player that precipitated the situation described in the paper: the publisher. In considering this essay and reading the exchange between Clark et al and Munday et al, I found it shocking that the editors and Nature and the reviewers involved in peer-review efforts let the situation unfold as it did. For example, in a short essay on its website titled "Challenges in irreproducible research,"

(https://www.nature.com/collections/wjsrmrdnsm) Nature states "No research paper can ever be considered to be the final word" (a parallel of the sentence in Williamson et al's line 104), though this is what Nature seems to have declared when publishing an article by Clark et al titled "Ocean acidification does not impair the behaviour of coral reef fishes". Why did editors and peer-reviewers not balk at this? Exploring this question would give the Williamson et al essay greater relevance within the literature. The review by Dupont makes a good point when it suggests expanding the Williamson et al essay to include another controversy that played out in Nature.

Along a related line of thought, the essay also omits consideration of why scientists like Clark et al and journals like Nature might have chosen to frame their work as they did and the implications their actions have on public trust of scientific information. Exciting and controversial articles in high-profile journals reward the authors and journals with media attention and higher "scores" in algorithms that aim to characterize prominence. Neither of these metrics denotes quality or trustworthiness, two characterizes that are vital for the public to trust scientific information and advice related to carbon dioxide emissions. The Williamson et al essay is important in setting boundaries for acceptable behavior in research science, but, as it is currently written, it does not touch on the larger consequences of the unfortunate actions it focuses on.

3) I would like to see the authors outline what would have been a better path for the Clark et al team at each step of the process of their work (initiation during grant writing, designing, interpreting) to avoid the situation that unfolded. Text on this idea could also include discussion of the value of institutional efforts in this process, for example the community building work done for US scientists by the Ocean Carbon Biogeochemistry program of the Woods Hole Oceanographic Institution.

4) Williamson et al bring up that the use of language in Clark et al deviated from typical academic literature in its boldness. I support Williamson et al's decision to convey the tone of the Clark et al arguments. Yet, they did not explicitly call out Clark et al for conveying information in an unemotional way, and I ask Williamson et al to consider if it would be valuable to do so. I acknowledge that emotions are not supposed to be included in papers on marine biology, but I see this Williamson et al paper as focused on the human actors doing science, which makes emotive language fair game for discussion.

I also ask Williamson et al to consider the value of their use of emotional language in their essay. I see objective language as more powerful than language that relies on emotive words to emphasize a point. For example, language like "unambiguously-titled" (line 39) made me chuckle, but may be a bit too cheeky to include. Reviewer 1 also remarked on this line "For the purpose of the appearance of objectivity, I recommend removing the phrase 'an unambiguously titled' and replacing it with the phrase 'the paper titled'. It allows the reader to draw their own conclusions about the Clark et al. 2020a paper's title from the argument that you present below." In another example, language on lines 43-45 ("Since Clark et al. went to 'great lengths' to replicate earlier work yet failed to get the same results,") feels sarcastic to me as a reader, which I don't appreciate in this type of professional setting.

5) A minor point: on lines 28-29 the authors should consider using more current references to characterize the ocean acidification literature as Kroeker et al. 2013 and Wittmann and Pörtner 2013 are too old to include the vast majority of literature on species sensitivity to ocean acidification.

---

## Short Comment (SC4) · 21 Dec 2020

The paper of Williamson et al. highlights several interesting topics associated to the scientific work in Ocean Acidification (which could be easily extrapolated to other scientific areas), mainly: - On the highly variable responses of the organisms faced to environmental stressors, - On the impossibility of recreating the real world in laboratory conditions, and - On the importance of creating a positive atmosphere to discuss on the scientific work. Specially, on the limits of the experimental studies, mainly those in laboratory.

It is increasingly documented that the responses of marine animals to stressors such

as ocean acidification can vary widely among and within species (Duarte et al., 2015, Shaw et al., 2016). Differential tolerance to environmental stressors is partly due to the conditions to which the individuals have been naturally exposed (Vargas et al., 2017), so it is important to consider those conditions prior to the experiments in order to correctly understand the consequences of Ocean Acidification. Given that it is impossible to determine all the abiotic conditions to which species are exposed to in nature, as Williamson et al. suggest, it is impossible to recreate them in mesocosms. In addition, animal condition (fitness) could vary through conditioning and experimentation, and so it should be taken into consideration as well. All this calls for caution when the results of the laboratory experiments are interpreted and, more importantly, in line with Williamson et al., when they are compared with studies carried out in different areas or in different times in the same area, either with the same or with a different species. This doesn't mean that laboratory experiments are not a good tool to understand how nature works. In fact, laboratory experiments have proved to be key tools to understand, for example, how human activity (e.g. Ocean Acidification) affects different ecosystems. This simply tries to highlight how difficult it is to compare the results of laboratory experiments and emphasize that they are normally complementary to each other and by no means the final truth in the generation of new scientific research. Williamson et al., describe clearly these points and, in this context, their article contributes to the discussion and understanding of the usefulness of laboratory experiments in ocean acidification research. Finally, as highlighted by Clements et al. (2020), it is not necessary to be harsh in the peer review process, and I would add that it is not necessary to be harsh after the review process either. That would allow the creation of an atmosphere to carry out this type of research (Williamson et al.,) which is, at the end of the day, what motivates everyone involved.

References

Clements, J.C. 2020. Don't be a prig in peer review. Nature. 585, 472. Duarte, C., Navarro, J.M., Acuña, K., Torres, R., Manríquez, P.H., Lardies, M.A., Vargas C.A.,

Lagos, N.A. & Aguilera, V. 2015. Intraspecific variability in the response of the edible mussel Mytilus chilensis (Hupe) to ocean acidification. Estuaries Coasts. 38:590–598 Shaw, E. C., Carpenter, R. C., Lantz, C. A. & Edmunds, P. J. 2016. Intraspecific variability in the response to ocean warming and acidification in the scleractinian coral Acropora pulchra. Marine Biology. 163, 210.

Cristian Duarte

---

## Author Comment (AC1) · 15 Jan 2021

We appreciate the constructive comments provided by Solan, with five grouped suggestions for additional content. Solan confirms that there are many fundamental methodological issues involved, affecting the full range of scientific process ‒ not just relating to experimental design, but also the analysis and interpretation of results. We fully recognise the importance of the issues raised for consideration. Comment-specific responses are provided here, and we also (briefly) cover several of these points in additional text of our manuscript.

COMMENT: 1. The opinion reinforces the argument about the need for proper experimental design, both within and across experiments, but doesn't explicitly provide solutions or minimal requirements. I have emailed you the recent Haddaway perspective (https://doi.org/10.1038/s41559-020-01295-x), maybe some of the styling from that would be good, but either a Boxed flow diagram or take home message at the end of each section of what readers should do would be good.

RESPONSE: Haddaway et al. (2020) address issues relating to (the lack of) rigour in systematic evidence syntheses in ecological disciplines, providing many key insights in that regard. We appreciate that their paper is primarily brought to our attention as an example of styling, with aspects that could be reflected in our manuscript. However, the approach and scope of Haddaway et al. (2020) is very much broader than our own. In particular, we need to avoid "mission creep" (one of the pitfalls they identify), by trying to change what was intended to be a focused commentary on one replicability case study into a much more comprehensive best-practice guide for ocean acidification, or even for experimental biology as a whole. For the former, we acknowledge that the overview guidance of a decade ago (Riebesell et al., 2011) now requires updating, to take account of methodological advances and greatly increased understanding during the past decade ‒ in particular, the increased emphasis on multifactorial studies (Boyd et al., 2018), rigour in experimental design (Cornwall and Hurd, 2016), and linkages between experimental studies and modelling (Ullah et al., 2020). Whilst our manuscript was clearly not meant to fill that gap, we now include the following additional text to show our recognition of the importance of such issues:

"Future ocean acidification experiments would also benefit from an updating of Riebesell et al. (2011), to provide improved best practice guidance on the key parameters that can affect results".

COMMENT: 2. Three things are missing for me. (i) one glove doesn't fit all, and what is high variability in one system (e.g. pelagic) is within the noise in another system (e.g. benthic), and that needs to be recognized, especially in review. Variability in carbonate chemistry between systems has not been summarised anywhere, although

could be now that there are so many papers, and then these need to be couched within temporal variation (e.g. diurnal) for the same system. (ii) the detection of treatment effects is valid, even if the carbonate chemistry is different to other experiments. You hint at this, but this would benefit from some elaboration. (iii) reading most ocean acidification papers, the narrative is that OA is 'bad' which is not necessarily true. I take the point that calcifiers are affected, but would challenge the statement that they are most sensitive as this is only one parameter and there is a bias in the literature (people have picked calcifiers). You highlight other examples, including behaviour, in non-calcifying species which could be equally devastating to that species. I key message is that this literature base needs to move on from documenting effects, and think about what the consequences of those effects are for species interactions, fitness, reproduction/growth and the rest of the foodweb etc etc.

RESPONSE: We recognise the importance of these three issues, and minor edits and additions to our manuscript have been made to indicate that awareness. For example:

"This [laboratory-based] approach has the advantage of enabling statistical testing of cause and effect for single factors, yet necessarily omits many of the complexities of natural conditions, that may involve temporal as well as biotic and abiotic environmental factors."

"Effects on the production of shells and skeletons have been a major research focus, but reduced calcification is not the only impact; there is also strong evidence for low pH affecting many other physiological processes (Pörtner et al., 2014; Baumann, 2019; Hurd et al., 2020), including vertebrate and invertebrate behaviour (Clements and Hunt, 2015; Cattano et al., 2018; Zlatkin and Heuer, 2019)."

But there is risk of being diverted, with discussion of any one of these topics (that each could justify separate monographs) resulting in our manuscript being very much longer than the "few pages" specified in Biogeosciences' guidance for an Ideas and Perspectives article. Even including brief mention could result in the criticism that our coverage

of such issues is superficial, not reflecting their importance, and should therefore be expanded.

COMMENT: 3. Is there a standard checklist, or could you provide one in this article, of what authors should be reporting in every paper, e.g. in a table in supplementary? The carbonate chemistry, but what else? Which of these are necessary, and which are nice to have?

RESPONSE: We recognise that the suggested checklist could be a key component of a comprehensive, updated good-practice guide ‒ and very much hope that the preparation of such material might be stimulated by the current discussions (as indicated by our response and manuscript edit already given above). However, the drafting of such a checklist is not a trivial issue; it will require thorough consideration of all the quantitative and qualitative issues involved, on a global basis. We therefore consider that such effort, whilst highly desirable, is outside the scope of our manuscript.

COMMENT: 4. It would be good if you could add some commentary about being pragmatic. Alkalinity, in particular, is expensive so an experiment with hundreds of replicates cannot hope to achieve regular samples from all units on a daily basis. You need some, but there are ways to achieve something sensible (e.g. a set number of aquaria within each treatment, once a week or something). There has to be some common sense, but also some indication of what the acceptable minimum is. The point I am making, and have witnessed at several OA meetings, is that the conversation about chemistry can go way further than is needed when discussing accuracy, repeatability and reproducibility. All of these depend on the system you are in, what the question you are asking is, and what is practically possible, i.e. the requirements are context dependent. A related point is that what works in one system should not dictate what is acceptable in another system. An analogy is the US water quality standards – when they were brought in some states were automatically above threshold as the ground composition was markedly different to the areas where the standards were formulated.

RESPONSE: We recognise the need for pragmatism in achieving an appropriate balance between perfectionism (controlling and measuring every conceivable influence, to three decimal places) and real-world resource limitations. We also recognise that chemists and biologists may have different priorities in that regard, and that the adopted approach for any replication study should be context dependent. Such ideas are covered in the following additional text:

"Since a very wide range of factors are potentially important, pragmatism will be needed with regard to associated issues of resource deployment and measurement accuracy, recognizing that chemists and biologists may have different priorities on such matters".

These issues are already covered, to some degree, in the existing best-practice guidance for ocean acidification studies (Riebesell et al., 2011). We consider that such guidance would benefit from updating, as already mentioned; nevertheless: i) inconsistencies in alkalinity measurements per se were not identified as a reason why Clark et al. (2020) did not find the same effects reported in previous studies; ii) the fluctuating and unstable pCO2 conditions in Clark et al. (2020) that were considered important did not require particularly sophisticated nor expensive techniques for their detection and control; and iii) differences in carbonate chemistry were only 1 out of 16 factors identified by Munday et al. (2020) as potentially influencing the outcome of the experiments.

COMMENT: 5. Variability – there are more sophisticated statistical methods available to look at variability and outliers. There is a danger that trying to make everyone confirm to a very controlled set of conditions means that you lose the insights from the variability that you have factored out. Part of the answer has to be embracing variability and using appropriate statistical approaches to account for and/or explore them. Meta-analysis is one way to look at multiple experiments, but not the only way and much could be done with mixed modelling, GLMM, GAMM and then specific analyses that analyse outliers (rather than account for them).

RESPONSE: We agree that environmental variability should be embraced, rather than factored out, and that there are opportunities for innovative methods (both statistical and model-based) to investigate its effects. The take-home message from our manuscript is intended to be fully consistent with that approach: there needs to be a spectrum of 'replication' experiments from those that are intended to be as similar as possible to those that are known to radically differ, with results interpreted accordingly. Acknowledgement of such a spectrum goes some way to resolving disputes on whether the conditions for a valid test of reproducibility have been met. There is also a second fundamental issue: the need for interpretation of any single study to take account of the wider body of relevant evidence.

REFERENCES CITED IN RESPONSES: Baumann, H.: Experimental assessments of marine species sensitivities to ocean acidification and co-stressors: how far have we come?. Can. J. Zoology, 97, 399-408, https://doi.org/10.1139/cjz-2018-0198, 2019.

Boyd, P.W., Collins, S., Dupont, S., Fabricius, K., Gattuso, J.-P., Havenhand, J., Hutchins, D.A., Riebesell, U., Rintoul, M.S., Vichi, M., Biswas, H., Ciotti, A., Gao, K., Gehlen, M., Hurd, C.L., Kurihara, H., McGraw, C.M., Navarro, J.M., Nilsson, G.E., Passow, U. and Pörtner, H.-O.: Experimental strategies to assess the biological ramifications of multiple drivers of global ocean change – A review. Glob. Change Biol., 24, 2239-2261, https://doi.org/10.1111/gcb.14102, 2018.

Cattano, C. J., Claudet, J., Domenici, P., and Milazzo, M.: Living in a high CO2 world: A global meta-analysis shows multiple trait-mediated fish responses to ocean acidification, Ecol. Monogr., 88, 320-335, https://doi.org/10.1002/ecm.1297, 2018.

Clark, T. D., Raby, G. D., Roche, D. G., Binning, S. A., Speers-Roesch, B., Jutfelt, F. and Sundin, J.: Ocean acidification does not impair the behaviour of coral reef fishes. Nature, 577, 370-375, https://doi.org/10.1038/s41586-019-1903-y, 2020.

Clements, J. C. and Hunt, H. L.: Marine animal behaviour in a high CO2 ocean, Mar. Ecol. Progr. Ser., 536, 259-279, https://doi.org/10.3354/meps11426, 2015.

Cornwall, C.E. and Hurd, C.L.: Experimental design in ocean acidification research: problems and solutions. ICES J. Mar. Sci., 73, 572-581, https://doi.org/10.1093/icesjms/fsv118, 2016.

Haddaway, N.R., Bethel, A., Dicks, L.V., Koricheva, J., Macura, B., Petrokofsky, G., Pullin, A.S., Savailaakso, S. and Stewart, G.B. : Eight problems with literature reviews and how to fix them. Nature Ecol. Ecol., 4, 1582-1589, https://doi.org/10.1038/s41559-020-01295-x, 2020.

Hurd, C.L., Beardall, J., Comeau, S., Cornwall, C.E., Havenhand, J.N., Munday, P.L., Parker, L.M., Raven, J.A., and McGraw, C.M.: Ocean acidification as a multiple driver: how interactions bewteeen changing seawater carbonate parameters affect marine life, Mar. Freshwater Res., 71, 263-274, https://doi.org/10.1071/MF19267, 2020.

Munday, P. L., Dixson, D. L., Welch, M. J., Chivers, D. P., Domenici, P., Grosell, M., Heuer, R. M., Jones, G. P., McCormick, M. I., Mark Meekan, M., Nilsson, G. E., Ravasi, T. and Watson, S.-A.: Methods matter in repeating ocean acidification studies. Nature (online), doi: 10.1038/s41586-020-2803-x, 2020.

Pörtner, H.-O., Karl, D. M., Boyd, P. W., Cheung, W. W. L., Lluch-Cota, S. E., Nojiri, Y., Schmidt, D. N., and Zavialov, P. O.: Ocean systems, in: Climate Change 2014: Impacts, Adaptation, and Vulnerability, Part A: Global and Sectoral Aspects. Contribution of Working Group II to the Fifth Assessment Report of the Intergovernmental Panel on Climate Change, edited by Field, C. B., Barros, V. R., Dokken, D. J. Mach, K. J., Mastrandrea, M. D., Bilir, T. E., Chatterjee, M., Ebi, K. L., Estrada, Y. O., Genova, R. C.,Girma, B., Kissel, E. S. Levy, A. N., MacCracken, S., Mastrandrea, P. R., and White, L. L., Cambridge University Press, Cambridge, UK, 411-484, 2014.

Riebesell, U., Fabry, V.J., Hansson, L. and Gattuso, J.-P.: Guide to Best Practices for Ocean Acidification Research and Data Reporting, European Commission, Brussels, http://dx.doi.org/10.2777/66906, 2011.
Ullah, H., Nagelkerken, I., Goldenberg, S.U. and Fordham, D.: Combining mesocosms with models to unravel the effects of global warming and ocean acidification on temperate marine ecosystems. EcoEvoRxiv Preprints (online); doi: 10.32942/osf.io/zs78v, 2020.

Zlatkin, R. L., and Heuer R. M.: Ocean acidification affects acid–base physiology and behaviour in a model invertebrate, the California sea hare (Aplysia californica), Roy. Soc. Open Sci., 6, 191041. http://dx.doi.org/10.1098/rsos.191041, 2019.

---

## Author Comment (AC2) · 15 Jan 2021

We welcome both the overview summary and the three specific suggestions provided by Dupont. Specific responses follow for the latter.

COMMENT: I would tone down the first sentence of the Abstract. I do not think that Clark et al. is addressing such a general question. They were rather focusing on the work by Munday and colleagues.

RESPONSE: We recognise that the first sentence of the Abstract ("Can experimental studies on the impacts of ocean acidification be trusted?") may seem over-general.

That wording had an informal genesis, arising from conversational comments made at an international science-policy meeting in early 2020; it also reflected online articles that used Clark et al. (2020a) as the basis for wider criticism of ocean acidification research (e.g. Starck, 2020). Furthermore, such wording is consistent with the claim by Clark (2017) that "Too many researchers make up or massage their data... only stringent demands for proof can stop them", without any indication that such criticism was specifically directed at Munday's research team. Nevertheless, the above background was not included in the main text of our manuscript – and, since we do not consider it necessary to do so, we have responded to Dupont's concerns by "toning down" the first sentence. Thus we now say "behavioural impacts",' rather than just "'impacts".

COMMENT: I would make the point that NO ocean acidification experiment can anyway simulate what is happening in the real world. So nobody is completely right or wrong if the experiment is well conducted (no fraud or big flaws) and honest with its limitations. You cannot design an experiment that will include the complexity of the real work and the time scale. This is another argument for a combination of experiments using different approaches (to answer different questions and then different part of the puzzle).

RESPONSE: We fully agree. We have therefore added the following text to our Introduction to make clear that ocean acidification experiments are necessarily simplifications of the real world, with the implications identified by Dupont;

"Laboratory experiments have investigated the biological impacts of ocean acidification through a reductionist approach; i.e. conditions are deliberately simplified. This approach has the advantage of enabling statistical testing of cause and effect for single factors, yet necessarily omits many of the complexities of natural conditions, that may involve temporal as well as biotic and abiotic environmental factors".

COMMENT: The manuscript focusses on the Munday vs. Clark et al. recent disagreement. Maybe it would be beneficial to can also include another example. A similar

discussion on the relevance of experimental studies occurred after the publication of the paper of Cornwall & Hurd (2015) highlighting the use of suboptimal experimental practices in many published articles in the field of ocean acidification. The article was covered by a short text in Nature entitled: 'Seawater studies come up short' (Nature 524, 18– 19; 2015) and followed by several media articles concluding that ocean acidification research was flawed. In a short response to Nature ("Laboratory seawater studies are justified", Hurd et al. 2015), we made similar points as this manuscript and highlighted the fact that laboratory studies are not ideal but one of the many tools (together with paleo studies, field work, models) that allow to capture the biological impact of ocean acidification.

RESPONSE: Dupont's reminder of previous discussions arising from Cornwall & Hurd (2015) is appreciated. This reference is now cited, in the context of recognising the need for appropriate rigour in the design and reporting of experimental studies. The issues raised in subsequent coverage by Nature were, however, somewhat different. Whilst Clark et al (2020a) considered that "small sample sizes and other methodological or analytical weaknesses" might account for minor impacts of ocean acidification previously reported, they dismissed that explanation for other studies showing large effects with large sample sizes. Instead, Clark (2017) and Clark et al. (2020a) emphasised the importance of video recording for their own studies, with implications of inadvertent bias or deliberate misconduct by others. When details of differences in experimental conditions were identified by Munday et al. (2020), there was only limited recognition by Clark et al. (2020b) that these differences could be relevant.

REFERENCES CITED IN RESPONSES

Clark, T. D.: Science, lies and video-taped experiments. Nature, 542, 139, https://doi.org/10.1038/542139a, 2017.

Clark, T. D., Raby, G. D., Roche, D. G., Binning, S. A., Speers-Roesch, B., Jutfelt, F. and Sundin, J.: Ocean acidification does not impair the behaviour of coral reef fishes.

Nature, 577, 370-375, https://doi.org/10.1038/s41586-019-1903-y, 2020a.

Clark, T.D., Raby, G.D., Roche, D. G., Binning, S. A., Speers-Roesch, B., Jutfelt, F. and Sundin, J.: Reply to: Methods matter in repeating ocean acidification studies, Nature (online), https://doi.org/10.1038/s41586-020-2803-x, 2020b.

Cornwall, C.E. and Hurd, C.L.: Experimental design in ocean acidification research: problems and solutions. ICES J. Mar. Sci., 73, 572-581, https://doi.org/10.1093/icesjms/fsv118, 2016.

Munday, P. L., Dixson, D. L., Welch, M. J., Chivers, D. P., Domenici, P., Grosell, M., Heuer, R. M., Jones, G. P., McCormick, M. I., Mark Meekan, M., Nilsson, G. E., Ravasi, T. and Watson, S.-A.: Methods matter in repeating ocean acidification studies. Nature (online), doi: 10.1038/s41586-020-2803-x, 2020.

Starck, W.: The tip of a JCU's junk science iceberg. Quadrant Online, https://quadrant.org.au/opinion/doomed-planet/2020/02/the-tip-of-a-jcus-junk-science-iceberg/, 2020.
* * *

---

## Author Comment (AC3) · 15 Jan 2021

The supportive comments by McCoy are appreciated. Whilst they do not seem to include specific issues requiring responses or text corrections (and are therefore not re-included here), they helpfully draw out two key messages. First, that the occurrence of different or controversial results from replication experiments reveal gaps in our understanding; second, that consensus science is what matters as the "big picture", with single studies providing the surrounding confidence envelope. We appreciate those insights, and now reflect them in the following revised and additional text:

". . . any single study does not disprove the consensus, since broadening the concept

of replication has the clear corollary that novel outcomes need to be interpreted using all available lines of evidence, with awareness of both similarities and differences in relation to previous work".

"Under what conditions should conclusions derived from one study be considered applicable (generalizable) to another, therefore enabling the underlying hypothesis to be tested, and potentially disproved, by the latter? The scientific benefits of that framing are greatest when the outcome of a replicability test is accepted by two research groups that initially favour different hypotheses - thereby requiring a more nuanced, non-confrontational framework for experimental planning, analysis and interpretation (Fanelli, 2018; Nosek and Errington, 2020a,b)".

REFERENCES CITED IN RESPONSE:

Fanelli, D.: Is science really facing a reproducibility crisis, and do we need it to? Proc. Natl. Acad. Sci. USA, 115, 2628-2631, https://doi.org/10.1073/pnas.1708272114, 2018.

Nosek, B. A., and Errington, T. M.: What is replication? PLoS Biol. 18, e3000691, https://doi.org/10.1371/journal.pbio.3000691, 2020a.

Nosek, B. A., and Errington, T. M.: Argue about what a replication means before you do it, Nature, 583, 518-520, https://doi.org/10.1038/d41586-020-02142-6, 2020b.

---

## Author Comment (AC4) · 15 Jan 2021

We greatly appreciate the constructive and supportive comments provided by Duarte. We consider that these comments provide a fair and accurate assessment of the importance of response variability in ocean acidification research, and such how this can be best be addressed and interpreted in experimental studies. We also endorse Duarte's concluding call for collaborative effort. Whilst we intend that our revised manuscript should reflect these views, we do not identify any criticisms requiring specific responses.

---

## Author Comment (AC5) · 15 Jan 2021

We thank RC1 for constructive general comments and for careful identification of ∼30 specific issues where improvements to the manuscript might be made. Our responses follow.

COMMENT: . . . I think that adding in a table or figure on this topic [criteria that should be used in replication versus reproducibility studies, and the implications for each] to help authors in the future to ensure that they meet these criteria in their studies would be helpful.

RESPONSE: We welcome this suggestion, and have added Figure 1 (appended here) to summarise the issues relating to the match between original and replication studies, and their implications. This is for the purpose of illustrating general principles rather than identifying specific criteria to separate the concepts of replication and reproducibility studies.

COMMENT: I think that the authors should consider editing the article in its entirety to prevent using subjective language throughout. I think that this topic is important and the audience as a whole will be more receptive if the authors' suggested ideas are laid out objectively and without subjective criticism of the published work on both sides.

RESPONSE: We welcome the explicit raising of issues relating to subjectivity and objectivity, recognizing the core importance of the latter in science. In response, we have made several edits to our manuscript, to make it more "neutral" in its language. Yet we are also aware of the following:

i) There is a timeless conundrum relating to absolute and relative truth: what is perceived as objective by one individual may be considered subjective by another. Cultural issues, semiotics (value-linked associations) and semantics (differences in word usage and intended meaning) can all affect the philological separation of objectivity and subjectivity.

ii) The Ideas and Perspectives format provides the opportunity to explore wider considerations relating to interpretation, opinion and framing, not being limited to the (objective) presentation and discussion of factual information.

iii) The main protagonist in this debate has had several high-profile opportunities for (subjective) communication of his opinions (e.g. Clark et al., 2016; Clark, 2017, 2020c), without the scrutiny of peer review.

We defend our own objectivity on a three-way basis: the lack of any collaborative engagement or other conflict of interests with the research groups that are in dispute;

our wider involvement in national and international research assessments, evaluations and reviews; and the different disciplinary routes by which we individually acquired ocean acidification expertise.

COMMENT: Lines 17-19: Is this the authors' argument? That the replication study was "confrontational"? It seems rather subjective and less constructive than the remainder of the abstract. Rather, it would be useful to provide an assessment of how to interpret and take further actions to understand the discrepancy when they arise from different experimental conditions here to close out the abstract. Below, you provide some really interesting suggestions for criteria for categorizing replication vs reproducibility studies, and I think a sentence along those lines would be more useful here.

RESPONSE: RC1 identifies the term "confrontational" as an example of unhelpful subjectivity. We disagree: the word was carefully chosen as a factual description of the approach taken by Clark (2017, 2020) and Clark et al. (2020a,b). Being confrontational is not necessarily undesirable; it certain circumstances, such action may be fully appropriate. But in this case we consider that it has been an obstacle, not an aid, to scientific progress.

COMMENT: Line 22: Add in an "i.e." in front of "increased"

RESPONSE: Accepted.

COMMENT: Line 26: Change "very well understood" to "high studied"

RESPONSE: There is a substantive difference in meaning between our text ("the chemistry of the carbonate system has been very well understood for decades") and the proposed edit ("the chemistry of the carbonate system has been high[ly] studied for decades"). Presumably the reason for the proposed change is that RC1 considers that the chemistry is not "very well understood", and therefore chemical aspects of ocean acidification remain uncertain. We recognize that pH-related chemical interactions can be complex (e.g. relating to use of buffers, calibrations, constants and pH scales), and

have therefore deleted "very". Nevertheless, we consider that it is important to point out that our knowledge of carbonate chemistry is much more complete than for the biological impacts of ocean acidification. Whilst this difference partly arises from the time period that the two systems have been investigated, more fundamental causes of complexity are also involved, affecting the predictability of system behaviours.

COMMENT: Line 35: Past studies have frequently used conspecific chemical alarm cues as the stimulus in these flume studies, and thus this stimulus should also be noted in the parentheses (such as in Welch et al. 2014 and Heuer et al. 2016). [Also Line 35]: I would recommend adding some examples of references here that have used this methodology so that interested readers can further read studies that have employed this methodology.

RESPONSE: We have changed "(e.g. from predators)" to "(from predators or conspecific alarm cues)", and have provided the following additional background:

"Several versions of such experimental conditions and treatments have been developed, with differences between protocols known to affect the strength of the response change (Jutfelt et al., 2017)".

The cited reference provides a review of this methodology that may be of interest to some readers.

COMMENT: Line 37: Again, provide some references here for examples when this methodology has been employed.

RESPONSE: Covered by above.

COMMENT: Line 39: For the purpose of the appearance of objectivity, I recommend removing the phrase "an unambiguously titled" and replacing it with the phrase "the paper titled". It allows the reader to draw their own conclusions about the Clark et al. 2020a paper's title from the argument that you present below.

RESPONSE: We consider the descriptor "unambiguously" to be objective, not subjective (nor pejorative). It draws attention to an important issue: the title of Clark et al. (2020a) is part of the problem. If it had been reworded "Ocean acidification does not always impair..." or even "Behaviour of coral reef fishes unaffected by ocean acidification" it would have been less strident in its assertion that this study invalidated previous work; i.e. less confrontational. Such a change of tone would have facilitated dialogue between the research groups directly involved - and may have obviated the need for this Ideas and Perspectives manuscript. The directness of the title of Clark et al (2020a) would seem to be deliberately provocative; it was reinforced by the accompanying press release by a co-author's institution (SLU, 2020) that stated "a new study, published in Nature, shows that ocean acidification does not affect the behaviour of coral reef fish at all".

COMMENT: Line 41: Replace "they" with "the authors"

RESPONSE: "The authors" could be used instead of "They". However, we do not consider that such a change significantly either improves clarity, removes potential misunderstanding or is relevant to the subjectivity/objectivity issue.

COMMENT: Line 43: Add in the year following Clark et al. (Clark et al. 2020a)

RESPONSE: For completeness, the date has been added.

COMMENT: Line 43.45: I recommend re-writing this statement in order to summarize the assertions in Clark et al. 2020a objectively. I recommend the following edit: "Since Clark et al. 2020 claim to have attempted to replicate the results obtained from earlier work without success, they imply that the earlier work was either unreliable, flawed, or fraudulent (Clark et al. 2017)."

RESPONSE: We have made minor edits to this sentence, as follows:

"Since Clark et al. (2020a) went to "great lengths" (in their own words) to replicate earlier work yet failed to observe the same effects, there was the implication that other researchers' work was flawed or even fraudulent. The latter interpretation is consistent

with earlier concerns expressed by Clark et al. (2016) and Clark (2017)".

We have retained "went to 'great lengths'" whilst now making clear that those words were used by Clark et al. (2020a) to assert that their experiments used, as far as possible, the same protocols and methods of previous studies. We therefore do not regard this as a subjectivity/objectivity issue.

COMMENT: Line 48: I recommend adding "substantial" here ("...attracted substantial media...").

RESPONSE: We do not consider "substantial" is justified. Whilst there was some media coverage, this was relatively specialised and mostly in the scientific press (e.g. Enserik 2020), without coverage by any major popular outlets (national press, TV or radio) as far as we are aware.

COMMENT: Line 48: I again recommend adding the year here ("... the apparent thoroughness of the approach described in Clark et al. (2020a)...").

RESPONSE: The date has been added.

COMMENT: Line 50: Please be clear about WHO identified these potential weaknesses. Also, as detailed in the reply by Munday et al. 2020a Nature, there were actually many more criticisms than three, so you should be clear that there were SEVERAL criticisms, and that you are highlighting three in particular.

RESPONSE: The three potential weaknesses were identified by combining quoted comments by Munday, given in a report by Enserick (2020), and independently by Widdicombe and Williamson in the collation of expert comments by the Science Media Centre (2020). All those comments were, however, relatively informal and of a rapid-response nature; they were made well before the much more comprehensive response by Munday et al. (2020), giving many more criticisms. Such information is readily available from the cited references. We do not consider it appropriate to say that we are highlighting three criticisms out of many, since we focus in this paragraph on the early

response - when most of the comments were relatively favourable.

COMMENT: Line 63: I recommend editing to include the phrase "but not limited to" (as in, "...included (but not limited to) the following...") .

RESPONSE: The proposed edit is redundant: "including" has the meaning that what follows is not comprehensive, i.e. is not limited to the information given.

COMMENT: Line 89: I recommend editing to say "...avoid creating confusion..." instead of "...avoid contributing to the confusion...". The recent Clark et al. 2020a paper created some confusion about the generality of CO2-induced effects on behaviour rather than contributing to existing confusion.

RESPONSE: The confusion referred to here is not specific to the effects of ocean acid-ification on fish behaviour. Instead, it was intended to refer to the different usages and definitions that exist in different disciplines relating to replication/replicability, repeata-bility and reproducibility. To clarify, the following re-wording is now used:

"This broad definition [of replication] has merit, although consistency is needed across all disciplines . . . to avoid contributing to terminological confusion in a contested topic area".

COMMENT: Line 90-91: I suggest editing this sentence to say: "First, it is important that replication studies examine key components of the original hypotheses, such as the life-stage dependence of the response to altered CO2 conditions."

RESPONSE: We have made minor edits to improve the clarity of this sentence. It is now closely similar to the suggested re-wording, as follows:

"First, it is important that the design of replication studies adequately addresses all key components of existing hypotheses; for example, the strong life-stage dependence of the response to high CO2 conditions".

COMMENT: Line 91-93: Your point in this sentence is not entirely clear. I believe

you are trying to say that all available evidence must be considered when evaluating potential controversies. Is that right? If so, please re-word so that aim is clear.

RESPONSE: We have re-wording this sentence to improve its clarity, as follows:

". . . any single study does not disprove the consensus, since broadening the concept of replication has the clear corollary that novel outcomes need to be interpreted using all available lines of evidence, with awareness of both similarities and differences in relation to previous work".

COMMENT: Line 94: For non-experts on this topic, provide some context for how studies of sensory physiology link to the studies on behaviour, which are the focus here.

RESPONSE: We now say: "sensory physiology (and hence behaviour)". Whilst additional details of the linkage could be provided, we do not consider this to be necessary.

COMMENT: Line 94: I believe that you mean Table 1 in Munday et al. 2020a, not supplementary table 1. Is that correct? If so, please correct this in the text.

RESPONSE: The reference to Supplementary Table 1 is correct: it can be accessed via the link given under Supplementary Information in the online version of Munday et al (2020). That information was compiled in January 2020, and did not include any papers later than 2019. Based on searches for the key words "fish, behaviour, acidification", we have now included data on five additional 2020 publications.

COMMENT: Line 95-96: For the sake of readability, I recommend using parentheses instead of commas around the phrase summarizing the researchers, institutions, and countries involved.

RESPONSE: Accepted.

COMMENT: Line 107: Add in a period after et al.

RESPONSE: Accepted.

COMMENT: Line 107-109: I recommend rewriting this sentence to the following: "Given the plethora of independent evidence, ocean acidification likely does have adverse impacts on fish behaviour. However, the resilience of fishes to altered CO2 is likely to vary depending on the species and circumstances under investigation."

RESPONSE: We have re-worded as follows:

"An objective summary of the global evidence is that ocean acidification can adversely affect fish behaviour under experimental conditions, whilst also recognising that the occurrence and scale of such impacts vary with circumstances and the species tested".

COMMENT: Line 112: Add in a comma between ecosystems and which.

RESPONSE: This sentence has been re-worded; comment no longer applicable.

COMMENT: Line 113: Change good to strong.

RESPONSE: This edit is not considered necessary.

COMMENT: Line 114: Rewrite this sentence as follows: "Our increasing appreciation for the complex relationship between ocean acidification and the ocean's biochemical, physiological, behavioural and ecological interactions are both scientifically exciting and sobering..."

RESPONSE: Sentence now re-worded, as follows:

"This increasing appreciation of the interactions between ocean acidification and other biochemical, physiological, behavioural, ecological and physical processes is both scientifically exciting and sobering...".

COMMENT: Line 116-119: These two sentences are somewhat difficult to understand. Can you re-write to improve their clarity and brevity? I believe that your point here is that the complexity of the relationship between ocean acidification and natural processes is to be expected given the variability that these species would have experienced throughout their evolutionary history. If so, just say that.

RESPONSE: Sentence now re-worded, as follows:

"The complexity of these interactions should, however, not be surprising, since marine species will have experienced natural variability in pH and $CO_2$ levels throughout their evolution. Species will have found different ways of responding, and response differences can therefore be expected to occur in experimental studies".

COMMENT: Line 130: The terms over-simplistic and unscientific are perhaps not the most constructive adjectives to use here. Consider changing this sentence to something like the following: "Given this known variability, the results from any one ocean acidification study will therefore be unable to overshadow all of the other previous findings from that area of research."

RESPONSE: Sentence now re-worded, as follows:

"Given this known variability, the results from any single ocean acidification study cannot provide the final word, over-riding other findings".

COMMENT: Line 131-132: I am not entirely clear what you are trying to assert with this sentence. My initial reading of this paragraph is that this sentence can be deleted in its entirety, to instead focus on how practices in the field can be improved generally, through more widespread publication of replication studies and negative results, which are notoriously difficult to publish and often end up in low impact journals, causing researchers not to prioritize dissemination of their important results.

RESPONSE: We have re-worked this paragraph, and have also included a 'Wider Implications' section to give more emphasis on how practices can be improved - whilst still noting the importance of publishing negative results.

COMMENT: Line 135-139: I am not entirely clear what you are suggesting here as an alternative approach. You suggest that the published back and forth between Clark et al. and Munday et al. is ineffective, but do not clearly layout an alternative framework for these sorts of controversies. What do you mean specifically by a "more nuanced,

nonconfrontational framework"? These specifics would be useful for taking constructive steps in the future.

RESPONSE: We have now expanded and explained these ideas in a new Wider Implications section, provided below (this text provides the context for Fig 1, as appended and already mentioned in response to an earlier comment):

"5. Wider implications

"The concept of generalizability (Nosek and Errington, 2020a) would seem crucial to the broader debate on replication. Under what conditions should conclusions derived from one study be considered applicable (generalizable) to another, therefore enabling the underlying hypothesis to be tested, and potentially disproved, by the latter? The scientific benefits of that framing are greatest when the outcome of a replicability test is accepted by two research groups that initially favour different hypotheses - thereby requiring a more nuanced, non-confrontational framework for experimental planning, analysis and interpretation (Fanelli, 2018; Nosek and Errington, 2020a,b).

[Figure 1 here]

"Figure 1 provides a diagrammatic summary of these issues, with situation (a) showing close congruence between two experimental studies, carried out by two research groups. If that very close match is recognised by both groups when Study #2 is planned (following the arrangements proposed by Nosek and Errington, 2020b), the replication provides a valid test of any hypotheses arising from Study #1. In contrast, situation (b) shows a pair of studies that only partly overlap, i.e. they differ in many regards, and where prior agreement between research groups on their congruence may not have been achieved. If results from both studies in situation (b) are consistent, the generalizability of Study #1 is extended. However, if inconsistent, the generalizability of Study #1 and Study #2 will each be constrained to its specific experimental conditions, with evidence from other studies providing the context for interpretation of the different outcomes. A range of intermediate situations between (a) and (b) can also occur.

"The above proposals for clearer "rules of engagement" for future replication studies could be greatly encouraged if research funders not only recognized that major insights can arise from closely similar or repeated work, but also required liaison between competing research teams as a condition of award in such circumstances. Our final recommendation is that high-profile publishers should give additional attention to the quality-control of potentially controversial papers, whilst also providing the opportunity for rapid, and preferably simultaneous, publication of responses by other researchers who may consider that their work has been unfairly criticized."

REFERENCES CITED IN RESPONSES

Clark, T. D.: Science, lies and video-taped experiments. Nature, 542, 139, https://doi.org/10.1038/542139a, 2017.

Clark, T.D.: Is ocean acidification driving fish bonkers? Nature Res. Ecol. Evol. Comm. (online), https://natureecoevocommunity.nature.com/posts/6519-is-ocean-acidification-driving-fish-bonkers, 2020.

Clark, T.D., Binning, S.A., Raby, G.D., Speers-Roesch, B., Sundin, J., Jutfelt, F. and Roche, D.G.: Scientific misconduct: the elephant in the lab. A response to Parker et al. Trends. Ecol. Evol. 31, 899-900, https://doi.org/10.1016/j.tree.2016.09.006, 2016,

Clark, T. D., Raby, G. D., Roche, D. G., Binning, S. A., Speers-Roesch, B., Jutfelt, F. and Sundin, J.: Ocean acidification does not impair the behaviour of coral reef fishes. Nature, 577, 370-375, https://doi.org/10.1038/s41586-019-1903-y, 2020a.

Clark, T.D., Raby, G.D., Roche, D. G., Binning, S. A., Speers-Roesch, B., Jutfelt, F. and Sundin, J.: Reply to: Methods matter in repeating ocean acidification studies, Nature (online), https://doi.org/10.1038/s41586-020-2803-x, 2020b.

Jutfelt, F., Sundin, J., Raby, G. D., Krång, A.- S. and Clark, T.D.: Two-current choice flumes for testing avoidance and preference in aquatic animals. Methods Ecol. Evol., 8, 379-390, https://doi.org/10.1111/2041-210X.12668, 2017.

[Figure]

Munday, P. L., Dixson, D. L., Welch, M. J., Chivers, D. P., Domenici, P., Grosell, M., Heuer, R. M., Jones, G. P., McCormick, M. I., Mark Meekan, M., Nilsson, G. E., Ravasi, T., and Watson, S.-A.: Methods matter in repeating ocean acidification studies, Nature (online), doi: 10.1038/s41586-020-2803-x, 2020.

Science Media Centre: Expert reaction to study looking at ocean acidification and coral reef fish behaviour. www.sciencemediacentre.org/expert-reaction-to-study-looking-at-ocean-acidification-and-fish-behaviour, 2020,

SLU: Ocean acidification is a big problem – but not for coral reef fish behaviour. SLU News/Swedish University of Agricultural Sciences, https://www.slu.se/en/ew-news/2020/1/ocean-acidification-a-big-problem–but-not-for-coral-reef-fish-behaviour, 2020.

[Figure]

[Figure]

**Fig. 1.** Figure 1: Visual summary of contrasting situations relating to (a) very close match-
ing and (b) part-matching of pairs of studies where Study #2 is intended to provide a test of
repeatability (and gene

---

## Author Comment (AC6) · 15 Jan 2021

We thank RC2 for supportive and constructive comments. We respond here to the issues raised for attention.

COMMENT: . . . the content of the Williamson et al. essay could be richer, which would help it appeal to a larger portion of the scientific community. The essay currently omits important aspects of the scientific process that led to the situation described and lacks concrete suggestions for how to avoid similar situations. The authors might also more carefully examine the language they used to avoid participating in a "toxic" exchange.

[Figure]

RESPONSE: We welcome the overall suggestion (also made by others) that the content of the manuscript should be broadened, and we have done do so - both within the existing text and in an additional, concluding section "Wider implications". As given below, with Figure 1 appended. Yet we are wary of trying to do too much, within an Ideas and Perspectives article that has to be limited to a few pages. Instead, we hope that this manuscript will stimulate further attention to the important issues raised regarding replication and reproducibility in general, and the more specific issues relating to ocean acidification.

"5. Wider implications

"The concept of generalizability (Nosek and Errington, 2020a) would seem crucial to the broader debate on replication. Under what conditions should conclusions derived from one study be considered applicable (generalizable) to another, therefore enabling the underlying hypothesis to be tested, and potentially disproved, by the latter? The scientific benefits of that framing are greatest when the outcome of a replicability test is accepted by two research groups that initially favour different hypotheses - thereby requiring a more nuanced, non-confrontational framework for experimental planning, analysis and interpretation (Fanelli, 2018; Nosek and Errington, 2020a,b).

[Figure 1 here]

"Figure 1 provides a diagrammatic summary of these issues, with situation (a) showing close congruence between two experimental studies, carried out by two research groups. If that very close match is recognised by both groups when Study #2 is planned (following the arrangements proposed by Nosek and Errington, 2020b), the replication provides a valid test of any hypotheses arising from Study #1. In contrast, situation (b) shows a pair of studies that only partly overlap, i.e. they differ in many regards, and where prior agreement between research groups on their congruence may not have been achieved. If results from both studies in situation (b) are consistent, the generalizability of Study #1 is extended. However, if inconsistent, the generalizability of

Study #1 and Study #2 will each be constrained to its specific experimental conditions, with evidence from other studies providing the context for interpretation of the different outcomes. A range of intermediate situations between (a) and (b) can also occur.

"The above proposals for clearer "rules of engagement" for future replication studies could be greatly encouraged if research funders not only recognized that major insights can arise from closely similar or repeated work, but also required liaison between competing research teams as a condition of award in such circumstances. Our final recommendation is that high-profile publishers should give additional attention to the quality-control of potentially controversial papers, whilst also providing the opportunity for rapid, and preferably simultaneous, publication of responses by other researchers who may consider that their work has been unfairly criticized".

COMMENT: The authors might also more carefully examine the language they used to avoid participating in a "toxic" exchange.

RESPONSE: We have carefully reviewed our choice of words, and have made several edits to ensure that our text is more neutral. We respond to specific issues below.

COMMENT: 1) This essay could strengthen its arguments by better incorporating the ideas presented in Nosek and Errington (2020)... this difference between failure and generalizability should be addressed in Williamson et al as it pertains to maintaining good norms of conduct in the field. Failure of a single study to support a hypothesis is a learning opportunity, not a reason to cast doubt on the rigor of prior work. A similar line of thinking is presented in the response by McCoy and in an Oceanography article by Busch, O'Donnell et al 2015 (http://dx.doi.org/10.5670/oceanog.2015.29).

RESPONSE: We appreciate these comments with associated recommendations, and have acted on them. Our submitted manuscript already cited Nosek and Errington (2020a,b), and more detailed discussion of their ideas is now given in the new Wider Implications section, as provided above. The reminder re the relevance of Busch et al. (2015) (now also cited) is also appreciated: those authors consider both reducible

and irreducible uncertainties, and the methods by which they might be addressed and communicated. Busch et al. also discuss the potential bias in meta-analyses arising from the non-publication of "no-effect" papers.

COMMENT: 2) The essay omits a major player that precipitated the situation described in the paper: the publisher. In considering this essay and reading the exchange between Clark et al and Munday et al, I found it shocking that the editors and Nature and the reviewers involved in peer-review efforts let the situation unfold as it did. For example, in a short essay on its website titled "Challenges in irreproducible research," (https://www.nature.com/collections/wjsrmrdnsm) Nature states "No research paper can ever be considered to be the final word" (a parallel of the sentence in Williamson et al's line 104), though this is what Nature seems to have declared when publishing an article by Clark et al titled "Ocean acidification does not impair the behaviour of coral reef fishes". Why did editors and peer-reviewers not balk at this? Exploring this question would give the Williamson et al essay greater relevance within the literature.

RESPONSE: We share these concerns, and (briefly) address the role of the publisher as a concluding remark in the Wider Implications section of.our revised manuscript, as given above. Whilst recognising that there are unavoidable imperfections and risks in the peer review process, there do seem to have been avoidable editorial failings with regard to Clark et al. (2020) and associated issues, including the long delay between the January submission of the Munday et al (2020) response and its October publication. We also point out here (but not in our manuscript) that Nature's policy regarding Correspondence currently excludes timely publication of comments relating to published material. In particular, a letter to the editor submitted in January 2020 by three of us, on the issues arising from Clark et al. (2020), was rejected as out of scope.

COMMENT: The review by Dupont makes a good point when it suggests expanding the Williamson et al essay to include another controversy that played out in Nature. Along a related line of thought, the essay also omits consideration of why scientists like Clark et al and journals like Nature might have chosen to frame their work as they did

and the implications their actions have on public trust of scientific information. Exciting and controversial articles in high-profile journals reward the authors and journals with media attention and higher "scores" in algorithms that aim to characterize prominence. Neither of these metrics denotes quality or trustworthiness, two characterizes that are vital for the public to trust scientific information and advice related to carbon dioxide emissions. The Williamson et al essay is important in setting boundaries for acceptable behavior in research science, but, as it is currently written, it does not touch on the larger consequences of the unfortunate actions it focuses on.

RESPONSE: The additional discussion of other controversies is tempting, yet could be diversionary. A separate response is given to Dupont to cover that specific suggestion. We recognise the importance of the wider issue of "impact", and its implications for quality and trustworthiness, yet have decided not to develop the current manuscript in that regard.

COMMENT: 3) I would like to see the authors outline what would have been a better path for the Clark et al team at each step of the process of their work (initiation during grant writing, designing, interpreting) to avoid the situation that unfolded. Text on this idea could also include discussion of the value of institutional efforts in this process, for example the community building work done for US scientists by the Ocean Carbon Biogeochemistry program of the Woods Hole Oceanographic Institution.

RESPONSE: The new, concluding section of our revised manuscript, as provided above in response to an earlier comment, gives (brief) attention to the practices that would enable more constructive replication studies. The ideas of Nosek and Errington (2020b) are highly relevant in that regard.

COMMENT: 4) Williamson et al bring up that the use of language in Clark et al deviated from typical academic literature in its boldness. I support Williamson et al's decision to convey the tone of the Clark et al arguments. Yet, they did not explicitly call out Clark et al for conveying information in an unemotional way, and I ask Williamson et

al to consider if it would be valuable to do so. I acknowledge that emotions are not supposed to be included in papers on marine biology, but I see this Williamson et al paper as focused on the human actors doing science, which makes emotive language fair game for discussion. I also ask Williamson et al to consider the value of their use of emotional language in their essay. I see objective language as more powerful than language that relies on emotive words to emphasize a point. For example, language like "unambiguously-titled" (line 39) made me chuckle, but may be a bit too cheeky to include. Reviewer 1 also remarked on this line "For the purpose of the appearance of objectivity, I recommend removing the phrase 'an unambiguously titled' and replacing it with the phrase 'the paper titled'. It allows the reader to draw their own conclusions about the Clark et al. 2020a paper's title from the argument that you present below." In another example, language on lines 43-45 ("Since Clark et al. went to 'great lengths' to replicate earlier work yet failed to get the same results,") feels sarcastic to me as a reader, which I don't appreciate in this type of professional setting.

RESPONSE: We consider that the use of non-technical, "direct English" generally improves communication. Thus there is no need to avoid boldness, in either a scientific or non-scientific context - provided that the words used are honest and appropriate; i.e. deliver the intended meaning, and are well-justified. What should be avoided is exaggeration or inaccuracy, not supported by evidence, or personal criticisms. Such failings may, however, become more apparent when the underlying message is more directly expressed. Whilst we recognise that the same words can be read in different ways, common usage should be the default. Thus we do not consider that "unambiguously", as used here, is an emotive word (cause for amusement or other strong reaction) - nor a subjective one, as regarded by the other Anonymous Reviewer. Similarly, the sentence "Clark et al. went to 'great lengths' to replicate earlier work yet failed to get the same results" was not intended to be sarcastic. Instead it stated, in those authors' own words (as we now make clear in the revised text), that they considered they had met all criteria for hypothesis-testing replication.

COMMENT: 5) A minor point: on lines 28-29 the authors should consider using more current references to characterize the ocean acidification literature as Kroeker et al. 2013 and Wittmann and Pörtner 2013 are too old to include the vast majority of literature on species sensitivity to ocean acidification.

RESPONSE: We agree that there is much more recent ocean acidification literature, and additional reviews are now cited in the Introduction. Nevertheless, those meta-analyses are still considered valid - and there have not been more recent attempts to provide systematic syntheses of a comprehensive nature, perhaps because of the magnitude of that task. Part of the point we wished to make in the Introduction (as scene-setting for later discussion) was that knowledge of response variability is well-established for ocean acidification experiments, and therefore provides the basic context for interpreting novel or unexpected results. References identifying important factors that are known to cause such response variability were identified later (lines 127-128) in our original manuscript.

REFERENCES CITED IN RESPONSES

Busch, D. S., O'Donnell, M. J., Hauri, C., Mach, K. J., Poach, M., Doney, S.c. and Signorini, S.R.: Understanding, characterising, and communicating responses to ocean acidification: challenges and uncertainties, Oceanogr., 28(2), 30-39, http://dx.doi.org/10.5670/oceanog.2016.29, 2016.

Clark, T. D., Raby, G. D., Roche, D. G., Binning, S. A., Speers-Roesch, B., Jutfelt, F. and Sundin, J.: Ocean acidification does not impair the behaviour of coral reef fishes. Nature, 577, 370-375, https://doi.org/10.1038/s41586-019-1903-y, 2020.

Munday, P. L., Dixson, D. L., Welch, M. J., Chivers, D. P., Domenici, P., Grosell, M., Heuer, R. M., Jones, G. P., McCormick, M. I., Mark Meekan, M., Nilsson, G. E., Ravasi, T. and Watson, S.-A.: Methods matter in repeating ocean acidification studies. Nature (online), doi: 10.1038/s41586-020-2803-x, 2020.

Nosek, B. A. and Errington, T. M.: What is replication? PLoS Biol. 18, e3000691, https://doi.org/10.1371/journal.pbio.3000691, 2020a.

Nosek, B. A. and Errington, T. M.: Argue about what a replication means before you do it. Nature, 583, 518-520, https://doi.org/10.1038/d41586-020-02142-6, 2020b.

―――――――――――――――――――――

(a)          (b)

Study          Study
**1             #2**

Other
studies

**1          #2**

**Fig. 1.** Visual summary of contrasting situations relating to (a) very close matching and (b) part-matching of pairs of studies where Study #2 is intended to provide a test of repeatability of Study #1

---

## Author Response (AR1)

Point-by-point responses to edits/comments from Biogeosciences' Handling Associate Editor, Tyler Cyronak on 15 January version of MS "Ideas and Perspectives: When ocean acidification experiments are not the same, reproducibility is not tested" (Williamson et al.)

**Edit/Comment:** Line 24. Delete 'primarily' and add 'currently' to sentence providing definition of ocean acidification. "*I would remove this* ["primarily"]. *I think it is pretty well established in the community that what we call 'ocean acidification' refers to the reduction in pH caused by anthropogenic CO2 emissions, while any other processes affecting pH are something different. In order to avoid any confusion with past geological changes that may have been caused by something other than CO2, I would add "currently" in front of "caused".*

**Response:** The original wording was closely based on the 2014 IPCC definition of ocean acidification (AR5 WGII Glossary). Rather than trying to make fine-tune changes to that definition, the sentence has now been shortened to: "Ocean acidification involves a reduction in seawater pH (increased hydrogen ion concentration), currently caused by increasing carbon dioxide ($CO_2$) in the atmosphere".

**Edit/Comment:** Line 29. Replace 'variable' by 'diverse'.

**Response:** Done.

**Edit/Comment:** Lines 36-37. "*I am usually not one to bring up my own citations in reviews, but I think this one by Lydia Kapsenberg and myself is particularly fitting here. It discusses the impact of carbonate chemistry variability under the framework of the 3-pronged concept of vulnerability put forth by Vargas et al. 2017. Kapsenberg, L. & Cyronak, T….*"

**Response:** Agreed that Kapsenberg & Cyronak (2019) is highly relevant here; citation added.

**Edit/Comment:** Line 48. Replace "and an intended replication" by "with the intention of replicating".

**Response**: Done.

**Edit/Comment:** Delete "unambiguously". *"Both anonymous reviewers comment on this word choice as being unnecessarily emotive. Speaking from being on the receiving end of a very emotive/non-objective comment in the literature (see Omega Myth discussions) I think it is important to maintain as much objectivity as possible. Overall I think you do a good job of that in this manuscript, while at the same time highlighting the somewhat provocative language…"* [remainder of comment not visible].

**Response:** The authors strongly wish to retain "unambiguously", since it is a factual and objective description. There is nothing inherently wrong (nor emotive) with being unambiguous; instead clarity is a desirable attribute, both in the titles and text of scientific papers, and also more widely. The problem only arises when a clearly-made statement is not well-supported by the evidence. It then becomes misleading.

**Edit/Comment:** Line 54. Insert "either".

**Response:** Done

**Edit/Comment:** Line 55.  Delete "the" and "is".

**Response:** The purpose of these edits is not fully clear.  The sentence has, however, been re-worded to end: "… either flawed or fraudulent, reflecting earlier concerns expressed by Clark et al. (2016) and Clark (2017)".

**Edit/Comment**: Line 61. "*Weird change in font size from here forward*".

**Response:**  Apologies.  This formatting error inadvertently occurred when the Word file was converted to pdf; final version will be more carefully checked.

**Edit/Comment:** Line 75.  Change "remedied" for "addressed".

**Response:** Done.

**Edit/Comment:**  Lines 105-106. "I would divide this into 2 sentences to make it easier for the reader".

**Response:**  Done

**Edit/Comment:** Line 108.  Delete "all"

**Response:** Done

**Edit/Comment:** line 109. Change "terminological" to "semantic".

**Response**: Done.

**Edit/Comment:** "*I wonder how much of this can be linked to observing effects vs "significant" effects. This brought the first figure in this Nature commentary to mind: https://www.nature.com/articles/d41586-019-00857-9...*"

**Response:**  Thank you for bringing Amrhein et al (2019) to our attention. The issue of over-interpreting a lack of statistical significance is an important one, and is now covered by a new sentence: "Second, the limitations of statistical analyses need to be recognized: statistically non-significant results do not necessarily mean there is no effect (Amrhein et al., 2019)".

**Edit/Comment:** Line 141. "*I think they will experience natural variability over many time/space scales beyond just evolution (see Kapsenberg GCB 2019).*"

**Response:** This sentence has been expanded to say ".. evolution and in their habitat diversity (Kapsenberg and Cyronak, 2019)".

**Edit/Comment:**  Line 141.  Change "have found" to "inherently have different vulnerabilities and"

**Response:**  Done.

**Edit/Comment:**  Line 151.  "Maybe replace by something like "a body of"

**Response:** Reworded to "over-riding the consensus of other findings".

**Edit/Comment:** Line 156.  Delete "best practice"

**Response**: Done.

**Edit/Comment:**  Line 161.  Change "would seem" to "seems"

**Response:** Done.

**Edit/Comment:** Line ~172.  "*Wrong word?"* [referring to "that"}

**Response:**  Sentence rephrased: "If both groups recognize that there is a.."

**Edit/Comment:**  Re Figure 1.  "*This is great, should you also put 'other studies' in the first panel?*".

**Response:**  Additional text added to Figure legend in response: "Whilst 'other studies' are also relevant to situation (a), their importance is increased when interpreting results from situation (b)".

**Edit/Comment:** Legend to Figure 1. Replace "pairs" by "pair".

**Response:**  We consider that the plural is correct since there are two pairs, one in (a) and the other in (b).

**Edit/Comment:** Lines 180-181.  "I like this idea".

**Response:**  No edit required

In addition to the changes discussed above, the authors have:

- Changed "reproducibility" to "repeatability" in the title of the paper, to make it more general
- Introduced the phrase "repeatability" in the text, to cover both reproducibility and replicability (Line 108: "Whilst both relate to the repeatability of outcomes, the test for reproducibility…")
- Thanked Tyler Cyronak in the Acknowledgements (with those thanked listed in alphabetic order).
- Included in the Reference List the citations for Amrhein et al. (2019) and Kapsenberg & Cyronak (2019), as suggested by Cyronak.  Also a new reference (Dahlke et al., 2018) as an example of differential vulnerability of different life-stages to OA impacts, potentially a major reason for replicability 'failure' in this case.

---

## Editor Decision (ED1)

**Ideas and Perspectives: When ocean acidification experiments are not the same, reproducibility is not tested**

Phillip Williamson[1], Hans-Otto Pörtner[2], Steve Widdicombe[3], Jean-Pierre Gattuso[4,5]

[1]School of Environmental Sciences, University of East Anglia, Norwich NR4 7TJ, UK

[2]Alfred Wegener Institute for Polar and Marine Research, 27515 Bremerhaven, Germany

[3]Plymouth Marine Laboratory, Plymouth, PL1 3DH, UK

[4]Sorbonne Université, CNRS, 06230 Villefranche-sur-mer, France

[5]Institute for Sustainable Development and International Relations, 75006 Paris, France

*Correspondence to*: Phillip Williamson (p.williamson@uea.ac.uk)

**Abstract**

Can experimental studies on the behavioural impacts of ocean acidification be trusted? That question was raised in early 2020 when a high-profile paper failed to corroborate previously-observed impacts on the responses of coral reef fish to high $CO_2$. New information on the methodologies used in the "replicated" studies now provides a plausible explanation: the experimental conditions were substantially different. High sensitivity to test conditions is characteristic of ocean acidification research; such response variability shows that effects are complex, interacting with many other factors. Open-minded assessment of all research results, both negative and positive, remains the best way to develop process-based understanding. As in other fields, replication studies in ocean acidification are most likely to contribute to scientific advancement when carried out in a spirit of collaboration rather than confrontation.

**1. Introduction**

Ocean acidification refers to a reduction in seawater pH (increased hydrogen ion concentration) over an extended period, typically decades or longer, caused  by increased uptake of carbon dioxide ($CO_2$) from the atmosphere. Associated chemical changes include increased concentration of bicarbonate ions and dissolved inorganic carbon, and reduction in the concentration of carbonate ions in the ocean and, unless compensated for, the body fluids of marine organisms. Although the chemistry of the carbonate system has been well understood for decades, research on the biological and ecological implications of anthropogenic ocean acidification only began in earnest about 20 years ago (Gattuso and Hansson, 2011). A wide range of potential consequences have since been identified, with an early appreciation of the  vulnerability of plant and animal species (Kroeker et al., 2013; Wittmann and Pörtner, 2013). Effects on the production of shells and skeletons have been a major research focus, but reduced calcification is not the only impact; there is also strong evidence for low pH affecting many other physiological processes (Pörtner et al., 2014; Baumann, 2019; Hurd et al., 2020), including vertebrate and invertebrate behaviour (Clements and Hunt, 2015; Cattano et al., 2018; Zlatkin and Heuer, 2019). Laboratory experiments have investigated the biological impacts of ocean acidification through a reductionist approach; i.e. conditions are deliberately simplified. This approach has the advantage of enabling statistical testing of cause and effect for single factors, yet necessarily omits many of the complexities of natural conditions, that may involve temporal as well as biotic and abiotic environmental factors.

**2. The challenge of contradictory results**

A two-step experiment has been used by many research groups to investigate the possible effects of ocean acidification on fish behaviour. Initially, individual fish are given a binary choice of water conditions in a flume tank, with one choice including an odour (e.g. from predators or a conspecific alarm cue) known to elicit an avoidance response. Those observations of discriminatory ability then provide the "control" strength of preference, for comparison with treatment results using the same choice under raised $CO_2$ (lowered pH) conditions throughout the test tank. Several versions of such experimental conditions and treatments have been developed, with differences between protocols known to affect the strength of the response change (Jutfelt et al., 2017).

Based on that binary-choice approach and an intended replication of previous work, Clark et al. (2020a) reported their findings in an unambiguously-titled paper: "Ocean acidification does not impair the behaviour of coral reef fishes". To exclude the possibility of inadvertent observer bias, they deployed video recording and automatic tracking software in their study, making that digital information openly available. They also used data simulations to conclude that previously reported results were "highly improbable", with an estimated likelihood of 0 out of 10,000 − assuming identical experimental conditions and that their own data were valid. Since Clark et al. (2020a) went to "great lengths" (in their own words) to replicate earlier work yet failed to observe the same effects, there was the implication that other researchers' work was flawed or even fraudulent. The latter interpretation is consistent with earlier concerns expressed by Clark et al. (2016) and Clark (2017).

In the context of a reported "crisis" in research reproducibility for many disciplines (Baker, 2016; Nature, 2018), Clark et al. (2020a) attracted media coverage and scientific responses, including praise for its thoroughness by several independent commentators (Enserinck, 2020; Science Media Centre, 2020). However, those initial reactions also identified three potential weaknesses. First, Clark et al. (2020a) *did* find several significant ocean acidification effects, contrary to the paper's title, although less dramatic than those previously reported. Second, their analysis gave scant attention to the extensive literature on factors causing variability in ocean acidification research. The third, more fundamental, concern related to how closely the original experiments had been repeated, and whether that issue had been thoroughly checked before the paper was published.

**3. Experimental differences**

Any deficiencies in the peer review of Clark et al. (2020a) were  nine months after its publication, with a detailed (online) critique by Munday et al. (2020a) that challenged the effectiveness of the claimed replication: "Clark et al. did not closely repeat previous studies, as they did not replicate key species, used different life stages and ecological histories, and changed methods in important ways that reduce the likelihood of detecting the effects of ocean acidification".

Experimental differences identified by Munday et al (2020a) between the original and repeated results included the following:

- Clark et al. (2020a) did not use clownfish, one of the original test species
- Adult and sub-adult fish were mostly used, rather than larvae and small juveniles (with older fish known to be less responsive to risk cues)
- For one species, the juveniles were from an inbred aquarium population (likely to be pre-adapted to high $CO_2$, and hence less sensitive)
- 85 Many experiments were carried out during a marine heatwave (with high temperatures known to reduce or reverse responses in the studied species)
- Dissolved $CO_2$ levels were unstable, with average daily $pCO_2$ range of 581 µatm in 2016 treatments. Such variability can reduce behavioural impacts (Jarrold et al., 2017) and did not match the stable conditions of directly-compared studies.

There were also crucial changes to the design of testing apparatus; the dilution and nature of odour cues; and the duration of tests. Such changes weakened the control response, hence reducing the likelihood of significant $CO_2$ treatment effects. In total, 16 differences between the original studies and the re-runs were described by Munday et al. (2020a), any one of which could potentially invalidate the comparisons.

The counter-argument, made at the time of the original publication (Enserink, 2020) and subsequently re-iterated by Clark et al. (2020b), is that minor experimental differences are inevitable, and can be considered as reflecting natural environmental variability. They should not matter if the original findings are widely applicable and robust. The question of what does or does not constitute a valid replication is therefore critical, yet inherently problematic. Since it is widely accepted that a fully-exact repeat of a biological study is impossible, due to the dynamic nature of both animate and inanimate factors ("No man ever steps in the same river twice; it is not the same man, nor is it the same river"; Heraclitus), it is important to distinguish "reproducibility" from "replicability". The test for reproducibility is conventionally limited to conditions where very tight control is achievable, e.g. in data treatments, or when re-using the original experimental set-up. In contrast, greater flexibility is allowed for testing replicability, reflected in a definition of replication as "a study for which any outcome would be considered diagnostic evidence about a claim from prior research" (Nosek & Errington, 2020a). This broad definition has merit, although consistency is needed across  disciplines (e.g. Stevens, 2017; Junk and Lyons, 2021) to avoid contributing to  confusion in a contested topic area.

Two further generic issues are also relevant here. First, it is important that the design of a replication study adequately addresses all key components of existing hypotheses; for example, the strong life-stage dependence of the response to high $CO_2$ conditions. Second, any single study does not disprove the consensus, since broadening the concept of replication has the clear corollary that novel outcomes need to be interpreted using all available lines of evidence, with awareness of both similarities and differences in relation to previous work. Munday et al. (2020a; Supplementary Table 1) identified 110 research papers published between 2009-2019 that investigated how ocean acidification might, or might not, affect the
behaviour and sensory physiology of fish. Out of 44 that involved coral reef fish, 41 of those studies (carried out by 68 researchers at 35 institutions in 15 countries), reporting significant effects, including several that used video recording, blind-testing, and raw-data publication. The remaining 66 papers (for other tropical, temperate and polar fish; marine and freshwater) provided additional support: 44 of those reported significant behavioural effects of ocean acidification. We are aware of five more recent publications on this topic, in addition to Clark et al. 2020a (Steckbauer et al., 2020; Rong et al., 2020; Jarrold et al., 2020; McIntosh, 2020; Roche et al., 2020); four of those reported significant effects.

A closely similar result was found in a meta-analysis of 95 marine and freshwater studies by Clements et al. (2020), with T.D. Clark included in the authorship team: they found that 64 of those papers reported either strong or weak behavioural effects. Whilst the proportion showing a strong effect declined over the period 2009 -2019, that decrease is unsurprising, since the early strong-effect studies were all on the most sensitive (marine) species. Additional independent evidence is provided by molecular studies, showing direct effects of high $CO_2$ on neurotransmission in fish (e.g. Schunter et al., 2019) and other taxa (e.g. Moya et al., 2016; Zlatkin and Heuer, 2019); further biochemical and pharmacological examples are
given by Munday et al. (2020a). An objective summary of the global evidence is that ocean acidification can adversely affect fish behaviour under experimental conditions, whilst also recognising that the occurrence and scale of such impacts vary with circumstances and the species tested.

**4. Taking account of response variability**

A recent IPCC assessment (Bindoff et al., 2019) confirmed the pervasive and complex effects of high $CO_2$ and warming, not
only on marine organisms and ecosystems but also on ecosystem services and society. Improved knowledge of all these response levels is crucial for effective mitigation and adaptation. This increasing appreciation of the interactions between
ocean acidification and other biochemical, physiological, behavioural, ecological and physical processes is both scientifically exciting and sobering, showing the difficulty in developing comprehensive understanding of this important component of ocean climate change. The complexity of these interactions should, however, not be surprising, since marine species will have experienced natural variability in pH and $CO_2$ levels throughout their evolution. Species will have found different ways of responding, and response differences can therefore be expected to occur in experimental studies.

Recognition of widespread response variability in ocean acidification experiments is not novel. It was noted for studies on survival, calcification, growth and reproduction in early meta-analyses (Kroeker et al., 2013), and subsequently provided the focus for much national and international research.  It is therefore now well-established that closely-related marine species can respond very differently to experimental pH treatments, and that the magnitude of single species responses can be affected by many factors. These influences include length of exposure; population-level genetic differences due to local adaptation; food availability; interactions with other stressors; seasonality; energy partitioning;and the sex of the organisms used in experiments (e.g. Thomsen et al., 2012; Suckling et al, 2014; Sunday et al., 2014; Breitburg et al, 2015; Vargas et al., 2017; Ellis et al., 2017); as well as physico-chemical conditions (Riebesell et al., 2011).

Given this known variability, the results from any single ocean acidification study cannot provide the final word, over-riding other findings.  Whilst many important uncertainties remain (Busch et al., 2016; Baumann, 2019; Hurd et al., 2020), we consider that scientific progress can be hindered by the sequence of polarising criticisms (Clark 2017, Clark et al 2020a), rebuttal (Munday et al 2020a), reply (Clarke et al.2020b) and a further point-by-point response (Munday et al. 2020b). A more constructive approach would  involve experimental co-design in a collaborative, comparative framework (Boyd et al., 2018), with appropriate rigour (Cornwall and Hurd, 2016) – that can still be consistent with scientific scepticism, replication tests and the reporting of negative results (Browman, 2016).  Future ocean acidification experiments would also benefit from an updating of Riebesell et al. (2011), to provide improved best practice guidance on the key parameters that can affect laboratory results.  Since a very wide range of factors are potentially important, pragmatism will be needed with regard to associated issues of resource deployment and measurement accuracy, recognizing that chemists and biologists may have different priorities on such matters.

**5.  Wider implications**

The concept of generalizability (Nosek and Errington, 2020a) would seem crucial to the broader debate on replication. Under what conditions should conclusions derived from one study be considered applicable (generalizable) to another, therefore enabling the underlying hypothesis to be tested, and potentially disproved, by the latter?  The scientific benefits of that framing are greatest when the outcome of a replicability test is accepted by two research groups that initially favour different hypotheses – thereby requiring a more nuanced, non-confrontational framework for experimental planning, analysis and interpretation (Fanelli, 2018; Nosek and Errington, 2020a,b).

Figure 1 provides a diagrammatic summary of these issues, with situation (a) showing close congruence between two experimental studies, carried out by two research groups. If  very close match is recognised by both groups when Study #2 is planned (following the arrangements proposed by Nosek and Errington, 2020b), the replication provides a valid test of any hypotheses arising from Study #1. In contrast, situation (b) shows a pair of studies that only partly overlap, i.e. they differ in many regards, and where prior agreement between research groups on their congruence may not have been achieved. If results from both studies in situation (b) are consistent, the generalizability if inconsistent, the generalizability of Study #1 and Study #2 will each be constrained with evidence from other situations between (a)

Figure 1: Visual summary of contrasting situations relating to (a) very close matching and (b) part-matching  of studies where Study #2 is intended to provide a test of repeatability (and generalizability) of Study #1. See text for more detailed explanation and discussion, including the importance of experimental co-design between research groups with contradictory hypotheses.

The above proposals for clearer "rules of engagement" for future replication studies could be greatly encouraged if research funders not only recognized that major insights can arise from closely similar or repeated work, but also required liaison between competing research teams as a condition of award in such circumstances. Our final recommendation is that high-profile publishers should give additional attention to the quality-control of potentially controversial papers, whilst also providing the opportunity for rapid, and preferably simultaneous, publication of responses by other researchers who may consider that their work has been unfairly criticized.

**Author contributions:** PW prepared the original draft with subsequent input and editing by all co-authors.

**Competing interests:** The authors declare that they have no conflicts of interest.

**Acknowledgements:** We are grateful for the constructive comments by Christian Duarte, Sam Dupont, Sophie McCoy, Martin Solan and two anonymous referees that have significantly improved this paper.

**References**

Baumann, H.: Experimental assessments of marine species sensitivities to ocean acidification and co-stressors: how far have we come? Can. J. Zool., 97, 399-408, https://doi.org/10.1139/cjz-2018-0198, 2019.

Bindoff, N. L., Cheung, W. W. L., Kairo, J. G., Arístegui, J., Guinder, V. A., Hallberg, R., Hilmi, N., Jiao, N., Karim, M. S., Levin, L., O'Donoghue, S., Purca Cuicapusa, S. R., Rinkevich, B., Suga, T. Tagliabue A., and Williamson, P.: Chapter 5. Changing ocean, marine ecosystems, and dependent communities, in: IPCC Special Report on the Ocean and Cryosphere in a Changing Climate, edited by Pörtner, H.-O., Roberts, D. C., Masson-Delmotte, V., Zhai, P., Tignor, M., Poloczanska, E., Mintenbeck, K.,Alegría, A., Nicolai, M., Okem, A., Petzold, J.,Rama, B., and Weyer, N. M., Intergovernmental Panel on Climate Change, https://www.ipcc.ch/srocc/chapter/chapter-5, 2019.

Boyd, P. W., Collins, S., Dupont, S., Fabricius, K., Gattuso, J.-P., Havenhand, J., Hutchins, D. A., Riebesell, U., Rintoul, M. S., Vichi, M., Biswas, H., Ciotti, A., Gao, K., Gehlen, M., Hurd, C. L., Kurihara, H., McGraw, C. M., Navarro, J. M., Nilsson, G. E., Passow, U., and Pörtner, H.-O.: Experimental strategies to assess the biological ramifications of multiple drivers of global ocean change – A review, Glob. Change Biol., 24, 2239-2261, https://doi.org/10.1111/gcb.14102, 2018.

Breitburg, D. L., Salisbury, J., Bernhard, J. M., Cai, W. J., Dupont, S., Doney, S. C., Kroeker, K. J., Levin, L. A., Long, W. C., Milke, L. M., and Miller, S. H.: And on top of all that… coping with ocean acidification in the midst of many stressors. Oceanogr., 28, 48-61, https://doi.org/10.5670/ oceanog.2015.31, 2015.

Browman, H. I.: Applying organized scepticism to ocean acidification research, ICES J. Mar. Sci., **73**, 529-536, https:/doi.org/10.1093/icesjms/fsw010, 2016.

Busch, D. S., O'Donnell, M. J., Hauri, C., Mach, K. J., Poach, M., Doney, S.C., and Signorini, S.R.: Understanding, characterising, and communicating responses to ocean acidification: challenges and uncertainties, Oceanogr., 28, 30-39, http://dx.doi.org/10.5670/oceanog.2016.29, 2016.

Cattano, C. J., Claudet, J., Domenici, P., and Milazzo, M.: Living in a high $CO_2$ world: A global meta-analysis shows multiple trait-mediated fish responses to ocean acidification, Ecol. Monogr., 88, 320-335, https://doi.org/10.1002/ecm.1297, 2018.

Clark, T. D.: Science, lies and video-taped experiments,Nature, 542, 139, https://doi.org/10.1038/542139a, 2017.

Clark, T.D., Binning, S.A., Raby, G.D., Speers-Roesch, B., Sundin, J., Jutfelt, F., and Roche, D.G.: Scientific misconduct: the elephant in the lab. A response to Parker et al., Trends. Ecol. Evol., 31, 899-900, https://doi.org/10.1016/j.tree.2016.09.006, 2016.

Clark, T. D., Raby, G. D., Roche, D. G., Binning, S. A., Speers-Roesch, B., Jutfelt, F., and Sundin, J.: Ocean acidification does not impair the behaviour of coral reef fishes, Nature, 577, 370-375, https://doi.org/10.1038/s41586-019-1903-y, 2020a.

Clark, T.D., Raby, G.D., Roche, D. G., Binning, S. A., Speers-Roesch, B., Jutfelt, F., and Sundin, J.: Reply to: Methods matter in repeating ocean acidification studies, Nature (online), https://doi.org/10.1038/s41586-020-2803-x, 2020b.

Clements, J. C. and Hunt, H. L.: Marine animal behaviour in a high $CO_2$ ocean, Mar. Ecol. Progr. Ser., 536, 259-279, https://doi.org/10.3354/meps11426, 2015.

Clements, J. C., Sundin, J., Clark, T. D., and Jutfelt, F. : An extreme decline effect in ocean acidification fish ecology, EcoEvoRxiv [preprint], https://doi.org/10.32942/osf.io/k9dby, 17 September 2020.

Ellis, R. P., Davison, W., Queirós, A. M., Kroeker, K. J., Calosi, P., Dupont, S., Spicer, J. I., Wilson, R. W., Widdicombe, S., and Urbina, M. A.: Does sex really matter? Explaining intraspecies variation in ocean acidification responses, Biol. Lett., 13, 20160761, https://doi.org/10.1098/rsbl.2016.0761, 2017.

Enserink, M.: Study disputes carbon dioxide-fish behavior link, Science, 367. 128-129, https://doi.org/10.1126/science.367.6474.128, 2020.

Fanelli, D.: Is science really facing a reproducibility crisis, and do we need it to? Proc. Natl. Acad. Sci. USA, 115, 2628-2631, https://doi.org/10.1073/pnas.1708272114, 2018.

Gattuso, J.-P., and Hansson, L. (Eds.): Ocean Acidification. Oxford University Press, Oxford, 326 pp, 2011.

Hurd, C.L., Beardall, J., Comeau, S., Cornwall, C.E., Havenhand, J.N., Munday, P.L., Parker, L.M., Raven, J.A., and McGraw, C.M.: Ocean acidification as a multiple driver: how interactions bewteeen changing seawater carbonate parameters affect marine life, Mar. Freshwater Res., 71, 263-274, https://doi.org/10.1071/MF19267, 2020.

Jarrold, M. D., Humphrey, C., McCormick, M. I., and Munday, P. L..: Diel $CO_2$ cycles reduce severity of behavioural abnormalities in coral reef fish under ocean acidification, Sci. Rep., 7, 10153, https://doi.org/10.1038/s41598-017-10378-y, 2017.

Jarrold, M. D., Welch, M. J., McMahon, S. J., McArley, T., Allan, B. J. M., Watson, S.-A., Parsons, D. M., Pether, S.M.J., Pope, S., Nicol, S., Smith, N., Herberet, N., and Munday, P. L.: Elevated $CO_2$ affects anxiety but not a range of other behaviours in juvenile yellowtail kingfish, Mar. Env. Res., 157, 104863, https://doi.org/10.1016/j.marenvres.2019.104863, 2020.

Junk, J. R., and Lyons, L.: Reproducibility and replication of experimental particle physics results, arXiv [preprint], arXiv:2009.06864(physics.data-an), 8 January 2021.

Jutfelt, F., Sundin, J., Raby, G. D., Krång, A.- S., and Clark, T.D.: Two-current choice flumes for testing avoidance and preference in aquatic animals. Methods Ecol. Evol., 8, 379-390, https://doi.org/10.1111/2041-210X.12668, 2017.

Kroeker, K. J., Kordas, R. L., Crim, R., Hendriks, I. E., Ramajo, L., Singh, G. S., Duarte, C. M., and Gattuso, J.-P.: Impacts
of ocean acidification on marine organisms: quantifying sensitivities and interaction with warming, Glob. Change Biol.,
19, 1884-1896, https://doi.org/10.1111/gcb.12179, 2013.

McIntosh, E. The Effect of Environmental Stressors on the Development and Behaviour of Larval *Oryzias latipes*, Honours
Thesis, University of Winnipeg, Winnipeg, Canada, http://hdl.handle.net/10680/1804, 2020.

Moya, A., Howes, E.L., Lacoue-Labarthe, T., Forêt, S., Hanna, B., Medina, M., Munday, P. L., Ong, J. S., Teyssié, J. L.,
Torda, G., Watson, S.-A., Miller D. J., Bijma J. & Gattuso J.-P.: Near-future pH conditions severely impact calcification,
metabolism and the nervous system in the pteropod *Heliconoides inflatus*, Glob. Change Biol., 22, 3888-3900,
https://doi.org/10.1111/gcb.13350, 2016.

Munday, P. L., Dixson, D. L., Welch, M. J., Chivers, D. P., Domenici, P., Grosell, M., Heuer, R. M., Jones, G. P.,
McCormick, M. I., Mark Meekan, M., Nilsson, G. E., Ravasi, T., and Watson, S.-A.: Methods matter in repeating ocean
acidification studies, Nature (online), doi: 10.1038/s41586-020-2803-x, 2020a.

Munday, P. L., Dixson, D. L., Welch, M. J., Chivers, D. P., Domenici, P., Grosell, M., Heuer, R. M., Jones, G. P.,
McCormick, M. I., Mark Meekan, M., Nilsson, G. E., Ravasi, T., and Watson, S.-A.: Additional response (online)

Nosek, B. A., and Errington, T. M.: Argue about what a replication means before you do it, Nature, 583, 518-520,

Na    https://doi.org/10.1038/d41586-020-02142-6, 2020b.

Pörtner, H.-O., Karl, D. M., Boyd, P. W., Cheung, W. W. L., Lluch-Cota, S. E., Nojiri, Y., Schmidt, D. N., and Zavialov, P.

Nc    O.: Ocean systems, in: Climate Change 2014: Impacts, Adaptation, and Vulnerability, Part A: Global and Sectoral
Aspects. Contribution of Working Group II to the Fifth Assessment Report of the Intergovernmental Panel on Climate
Change, edited by Field, C. B., Barros, V. R., Dokken, D. J. Mach, K. J., Mastrandrea, M. D., Bilir, T. E., Chatterjee, M.,
Ebi, K. L., Estrada, Y. O., Genova, R. C.,Girma, B., Kissel, E. S. Levy, A. N., MacCracken, S., Mastrandrea, P. R., and
White, L. L., Cambridge University Press, Cambridge, UK, 411-484, 2014.

Riebesell, U., Fabry, V. J., Hansson, L., and Gattuso, J.-P.: Guide to Best Practices for Ocean Acidification Research and
Data Reporting, European Commission, Brussels, http://dx.doi.org/10.2777/66906, 2011.

Roche, D. G., Amcoff, M., Morgan, R., Sundin, J., Andreassen, A. H., Finnøen, M. H., Lawrence, M. J., Henderson, E.,
Norin, T., Speers-Roesch, B., Brown, C., Clark, T.D., Bshary, R., Leung, B., Jutfelt, F., and Binning S.A.: Behavioural
lateralization in a detour test is not repeatable in fishes, Anim. Behav. 167, 55-64,
https://doi.org/10.1016/j.anbehav.2020.06.025, 2020

Rong, J., Tang, Y., Zha, S., Han, Y., Shi, W., and Liu, G.: Ocean acidification impedes gustation-mediated feeding behaviour by disrupting gustatory signal transduction in the black sea bream, *Acanthopagrus schlegelii,* Mar. Env. Res. 162, 195182, https://doi.org/10.1016/j.marenvres.2020.105182, 2020.

Schunter, C., Ravasi, T., Munday, P. L., and Nilsson, G. E.: Neural effects of elevated $CO_2$ in fish may be amplified by a vicious cycle, Cons. Physiol., 7, coz100, https://doi.org/10.1093/conphys/coz100, 2019.

Science Media Centre: Expert reaction to study looking at ocean acidification and coral reef fish behaviour, www.sciencemediacentre.org/expert-reaction-to-study-looking-at-ocean-acidification-and-fish-behaviour, last access 20

October 2020.

Steckbauer, A., Diaz-Gil, C., Alós, J., Catalán, I. A. and Duarte, C. M.: Predator avoidance in the European seabass after recovery from short-term hypoxia and different $CO_2$ conditions, Front. Mar. Sci. 5, 350, https://doi.org/10.3389/fmars.2018.00350, 2018.

Stevens, J. R.: Replicability and reproducibility in comparative psychology. Front. Psychol., 8, 862, https://doi.org/10.3389/fpsyg.2017.00862, 2017.

Suckling, C. C., Clark, M. S., Richard, J., Morley, S. A., Thorne, M. A., Harper, E. M., and Peck, L. S.: Adult acclimation to ambient temperature and pH stressors significantly enhances reproductive outcomes compared to short-term exposure, J. Anim. Ecol., 84, 773-784, https://doi.org/10.1111/1365-2656.12316, 2014.

Sunday, J. M., Calosi, P., Dupont, S., Munday, P. L., Stillman, J. H., and Reusch, T. B.: Evolution in an acidifying ocean. Trends Ecol. Evol., 29, 117-125, https://doi.org/10.1016/j.tree.2013.11.001, 2014.

Thomsen, J., Casties, I., Pansch, C., Körtzinger, A., and Melzner, F.: Food availability outweighs ocean acidification in juvenile *Mytilus edulis*: laboratory and field experiments. Glob. Change Biol., 19, 1017-1027, https://doi.org/10.1111/gcb.12109, 2012.

Vargas, C. A., Lagos, N. A., Lardies, M. A., Duarte, C., Manríquez, P. H., Aguilera, V. M., Broitman, B., Widdicombe, S., and Dupont, S.: Species-specific responses to ocean acidification should account for local adaptation and adaptive plasticity, Nature Ecol. Evol., 1, 1-7, https://doi.org/10.1038/s41559-017-0084, 2017.

Wittmann, A. C., and Pörtner, H. O.: Sensitivities of extant animal taxa to ocean acidification, Nature Clim. Change, 3, 995-1001, https://doi.org/10.1038/nclimate1982, 2013.

Zlatkin, R. L., and Heuer R. M.: Ocean acidification affects acid–base physiology and behaviour in a model invertebrate, the

California sea hare (*Aplysia californica*), Roy. Soc. Open Sci., 6, 191041. http://dx.doi.org/10.1098/rsos.191041, 2019.